# NLMOPTIMIZER: A NEUROSYMBOLIC FRAMEWORK AND BENCHMARK FOR OPERATIONS RESEARCH OPTIMIZATION PROBLEMS FROM NATURAL LANGUAGE

## ABSTRACT

Large Language Models (LLMs) are increasingly applied to structured reasoning tasks, but remain prone to generating outputs that are both syntactically coherent and semantically invalid, posing a serious challenge for the domain of mathematical optimization. In particular, applications to operations research (OR) problems, where problem descriptions are often ambiguous, context-rich, and semantically dense, are compromised by these issues and a dearth of publicly available datasets appropriately designed for both training and benchmarking model performance. In this paper, we address these issues by first introducing **NLMOptimizer**, a neurosymbolic framework built on two classes: (i) the **Problem** class, which systematically generates optimization problems; and (ii) the **SymInterchange** class, an exploratory suite of neurosymbolic methods intended to map word problems into structured, solver-executable forms. We then address the dearth of plausibly complex OR problems with the associated NLMOptimizer dataset, generated using **Problem**, which pairs structured natural-language descriptions with solver-checked mathematical programs across 1000 different linear (LP) and quadratic programs (QP) across integer, mixed-integer, and continuous types. We evaluate four instruction-tuned LLMs (LLaMa-3.3, LLaMa-4-Scout, Gemini-1.5-Pro, GPT-OSS-120B) under zero-shot prompting and observe substantial degradation on our dataset, with the strongest model dropping from 66.6% end-to-end accuracy on the NL4OPT benchmark dataset to 14.6% on NLMOptimizer. Our results indicate that (i) widely used benchmarks understate the difficulty of mapping natural language to formal OR optimization problem structure, (ii) current LLMs struggle to represent even modestly more complex OR optimization problems than LPs with three variables, and (iii) progress will require methods that directly target representational fidelity without training models to fit fixed examples.

## 1 INTRODUCTION

Despite remarkable advances in large language models (LLMs) and neural reasoning systems, today's artificial intelligence (AI) still struggles to integrate the depth and rigor of symbolic mathematics with the flexibility of natural language understanding. This gap is especially salient in mathematical optimization, where problem descriptions are often ambiguous, context-rich, and semantically dense, yet require precise formalization for solver execution. Operations research (OR), a branch of applied mathematics central to decision making in domains such as health care, logistics, and network security, exemplifies this challenge. In particular, optimization requires a faithful *problem representation* whose entities, relations, and algebraic constraints can be manipulated reliably. The Natural Language for Optimization (NL4OPT) challenge (Ramamonjison et al., 2022a), which seeks to translate text descriptions of OR problems into mathematical formulations, is thus both a high-stakes application and a difficult representation learning task. Recent systems such as **OptGen** (Ramamonjison et al, 2022), **OptiMUS** (AhmadiTeshnizi et al., 2024), and **ComplexOR** (Xiao et al., 2024) have demonstrated progress using intermediate representations, chain-of-thought prompting, and modular task decomposition. However, they remain fundamentally limited in their ability to ensure semantic alignment, manipulate symbolic structures, and scale to solver-executable outputs. Further, the dominant NL4OPT benchmark (Ramamonjison et al., 2022a) is restricted to

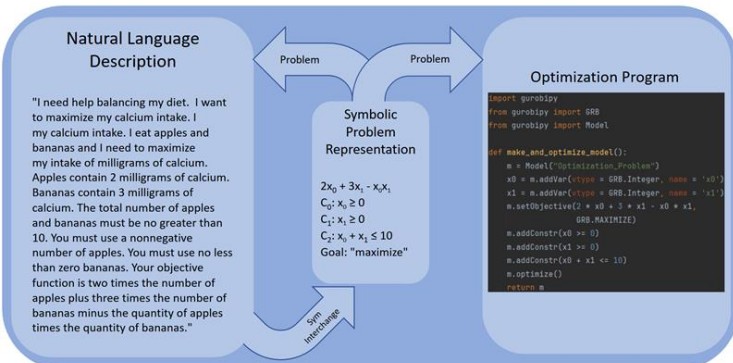

Figure 1: Design pattern of **Problem** and **SymInterchange** classes.

a small set of simple examples, failing to capture the diversity and structural richness of real-world OR problems (Xiao et al., 2024).

Most recently, Mostajabdaveh et al. (2025) introduced **ORQA**, a hand-crafted benchmark that asks models to identify objectives, constraints, entities, and interrelations from realistic textual cases. Despite strong instruction-tuned models, accuracies plateau well below expert performance and degrade when reasoning is elicited by chain-of-thought prompts.[1] We argue that such failures are not idiosyncrasies of prompting, but symptoms of a deeper representational deficit and highlight the need for neurosymbolic approaches that treat representation as first-class: not only recognizing patterns in text, but also learning to map across natural language, symbolic abstractions, and solver-ready code in a way that is precise, generalizable, and transparent. We argue that many practical optimization problems admit a compact description as semi-algebraic sets and exponential-polynomial expressions over *E-rings*, and that the proper challenge is reconstructing this from natural language descriptions.

We propose **NLMOptimizer**, a neurosymbolic framework that couples a scalable realistic OR optimization problem generator with a suite of methods to share and conform lightweight IR into executable code. Our contributions in this paper consist of:

1. **Problem generation at scale.** We introduce the **Problem** class, which generates optimization problems, symbolically represented as semi-algebraic varieties defined over exponential rings $(\mathbb{R}[x_1, \ldots, x_n])^{\exp}$ (Van Den Dries, 1984), and instantiate an initial dataset of 1000 linear and quadratic programs to enable head-to-head comparability with NL4OPT while increasing structural realism by including multiple resources types and larger variable counts.

2. **Empirical evaluation.** We benchmark four LLMs on both NL4OPT and 1000 **Problem**-generated examples. As with Mostajabdaveh et al. (2025), we observe that all models collapse under zero-shot prompting when faced with more realistic problems (e.g., continuous linear programs with 5+ variables), with Gemini-1.5-Pro End-to-End accuracy dropping from 66.6% (on NL4OPT) to 14.6% (on NLMOptimizer problems).

Together, these results demonstrate that current benchmarks dramatically underestimate task difficulty, that existing LLMs lack robust representations for optimization, and that principled symbolic interfaces are essential for advancing neurosymbolic reasoning and representation. While the total representational framework presented by **Problem** is scoped beyond polynomial terms, we restrict our empirical evaluation in this paper to LP and QP problems for its first release. This choice isolates and highlights representational difficulties without confounding it further by the introduction on non-standard polynomial objectives and constraints. This directly enables like-for-like comparison with NL4OPT, and avoids being able to dismiss observed degradations in performance due to significantly more challenging problems, as these degradations appear already at linear and quadratic cases with realistic scale and structure.

---

[1]See their Table 2 and human baseline, where the largest open model achieving 0.772 accuracy and a PhD baseline at 0.93, while CoT degrades with additional prompting.

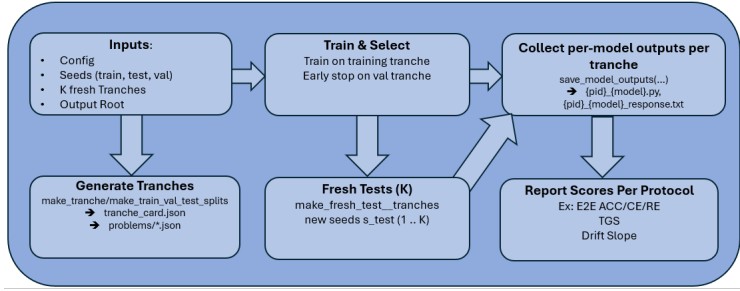

Figure 2: **NLMOptimizer** Protocol With **Problem** For Dynamic Tranche Evaluation

## 2 RELATED WORK AND PRELIMINARIES

We assess that fully automated end-to-end processes without human-in-the-loop interventions risk incorrect outputs, particularly for more complex, domain-specific real-world use cases. To motivate our proposed framework, we first review several LLM-based efforts and their related datasets that pursued complementary approaches to natural language modeling of optimization problems.

### 2.1 PRIOR BENCHMARKING EFFORTS FOR NL-BASED OPTIMIZATION MODELING

Developed for the Natural Language for Optimization competition, the NL4OPT dataset was developed to increase the accessibility and usability of optimization solvers by allowing general users to interface with them through natural language descriptions (Ramamonjison et al, 2022; Ramamonjison et al., 2022b), and it has served as the benchmark dataset for the NL4OPT task. This dataset presents a total of 1101 annotated Linear Programming (LP) word problems across 6 different domains created using the Prodigy tool to manually create and annotate 600 problems, with the remaining 501 samples and annotations were generated using a NER model. **OptiMUS** (AhmadiTeshnizi et al., 2024) is a natural language (NL) to optimization code generator that employs LLM-based agents in a multi-step process including problem specification, variable and formulae generation, code generation, and debugging. The authors evaluate their approach over both existing and new benchmark datasets, and demonstrate improvement over existing NL-based solver code generators. **ComplexOR** (Xiao et al., 2024) uses a Chain-of-Experts approach with multiple experts (or agents) with individual tasks and goals. These experts include a conductor, terminology interpreter, modeling expert, programmer, and evaluator. Work is orchestrated via conductor, and tasks pass bidirectionally between experts based on feedback from the modeling expert and evaluator agents. In addition to this novel approach, the authors also present a more challenging benchmark dataset that more closely resembles real-world optimization problems, although presently the formal 37 datasets mentioned in the paper have yet to be released for review. Recent results with **ORQA** (Mostajabdaveh et al., 2025) indicate that multiple-choice QA (MCQ) can diagnose gaps in *recognizing* optimization structure but underestimates the difficulty of *constructing* it: large models achieve respectable scores on reading-comprehension-like items yet fail sharply when entity–relation modeling is required, and chain-of-thought prompting often reduces accuracy. These outcomes align with a representation-centric view of LLM failures in OR. Prior datasets either rely on toy tasks, require code-generation and solver execution for scoring, or conflate notation bugs with modeling errors. **ORQA** improves on this by isolating model-component identification, but its MCQ format still cannot test constructive adequacy.

### 2.2 REPRESENTING AND SOLVING OPTIMIZATION PROBLEMS

An optimization problem captures decision-making under constraints with a goal to minimize, or dually, maximize a particular function. Constraints produce a set of options from which the decision-maker must choose. Whenever the set of options is empty, the problem is *infeasible*. For LLM systems that generate optimization routines, a symbolic formalism capturing the full spectrum of objective functions and constraint types is essential for expressivity and correctness.

Crucial for our purposes, semidefinite programming (SDP), a subclass of convex optimization problems where the feasible set is described by linear matrix inequalities, bridges the numerical algorithmic implementations of optimization and the core real algebraic geometric content. SDP captures non-linear algebraic constraints in a convex formulation, using auxiliary variables and moment representations, acting as the natural computational arena for embedding semialgebraic geometry into optimization solvers (Netzer, 2016). Formally, a program $\mathbf{P}$ is *semidefinite* if there are some $c \in \mathbb{R}^n$, and symmetric matrices $M_0, M_1, \ldots, M_n \in Sym_N(\mathbb{R})$ such that $\mathbf{P}$ is given by $\min \vec{c} \cdot \vec{a}$ subject to $M_0 + a_1 M_1 + \cdots a_n M_n \succeq 0$, where $M \succeq$ means that $M$ is a semipositive definite matrix.

Following standard model theory conventions, given a first order language $\mathcal{L}$, an $\mathcal{L}$-structure $\mathcal{M}$ is some domain $M$ whose elements satisfies a set of sentences in $\mathcal{L}$. For example, given $\mathcal{L} = \langle 0, 1, +, \cdot \rangle$, any field $F$ is defined over the language $\mathcal{L}$ using the field axioms (Marker, 2002). For a $D \subset M$, $D$ is a *definable set* if there is an $\mathcal{L}$-formula $\varphi$ such that $a \in D$ if and only if $\mathcal{M} \models \varphi(a)$. In theory, optimization problems rely on feasible sets, while in practice feasible sets and the objective functions are precisely those that are definable in terms from the first-order language of ordered commutative rings $\mathcal{L}_{or} = \langle 0, 1+, \cdot, < \rangle$. We can expand the language $\mathcal{L}_{or}$ to include a function symbol $E$, or $\exp$, and provide sentences that allow us to interpret $E$ as the standard real-valued exponential function. It is immediate that there is a bijective correspondence between the polynomials in any given ordered ring $R[X_1, \ldots, X_m]$ with indeterminates $X_i$, and the terms generated from $\mathcal{L}_{or}$ when adding the constants of $R$ to the $\mathcal{L}_{or}$ (without any extensions of the language, the terms are precisely in correspondence with polynomials in $\mathbb{Z}[X_1, \ldots, X_m]$) (Marker, 2002). A field is $F$ is said to be *real-closed* if there is a total order $<$ on F such that every positive element of $F$ has a square root in $F$ and every polynomial of odd degree with coefficients in $F$ has at least one root in $F$. The canonical real-closed field is the field of real numbers, $\mathbb{R}$. For any real-closed field $F$, a set $\mathcal{W} \subset F^n$ is said to be *basic closed semialgebraic set* if it is a finite intersection of sets defined by polynomial inequalities of the form $p_i(X_1, \ldots, X_n) \geq 0$ for $p_i \in F[X_1, \ldots, X_n]$, so that $\mathcal{W} = \{ \vec{x} \in F^n \mid \bigwedge_{i \in I_W}^r p_i(\vec{x}) \geq 0 \}$, while a general *semialgebraic set* (alternatively, *semialgebraic variety*) is a finite Boolean combination of basic closed algebraic sets (Netzer, 2016; Marker, 2002). Following the standard definition (see Van Den Dries (1984)), an E-ring $(R, E)$ augments a commutative ring $R$ with an exponential map $E : R \to R^\times$ satisfying $E(x + y) = E(x) \cdot E(y)$. E-polynomials are built inductively from $R$ by adjoining symbols closed under $+$, $\cdot$, and $E(\cdot)$.

## 3 METHODOLOGY

### 3.1 NLMOPTIMIZER

Our preliminary investigation concerns the representation of optimization problems, and our experiments principally concern benchmarking with the **Problem** class. We consider the challenge posed by correct intermediate representations (IR) as a matter to be addressed by future development of the **SymInterchange** class, detailed in Appendix C.3 We recognize following Section 2.2 that feasible sets are identifiable with terms satisfying relations determined by the elements of E-rings, we developed a class titled **Problem**, which randomly selects for terms $t :\equiv r \mid x \mid t + t \mid t * t \mid \exp(t)$, where $r \in \mathbb{R}$, are terms subject to the natural $\leq$ order relation on $\mathbb{R}$, subject to hyperparameter selection. Our pragmatic motivation for E-rings is summarized by Corollary B.11 (details in Appendix B).

**Corollary 3.1** (Pragmatic E-ring policy)**.** *If a natural language to intermediate representation mapping stays within $\mathbb{R}_{or}$, feasibility and optimality are* decidable*, admitting effective QE. If an NL $\to$ IR mapping stays within $\mathbb{R}_{an,\exp}$, we admit weak quantifier elimination.*

For our initial benchmarks, we set our generator to linear and quadratic programs, which are either classical programs, mixed integer, or integer programs, and have included a restricted implementation of the general problem class in Appendix C.1. This restriction is intentional: problems are guaranteed to admit QE by Cor. B.11, and it keeps algorithmic and solver complexity fixed while scaling representation complexity for the standard benchmark solver, Gurobi.[2] For our initial experiments, we generated a test batch of 1000 linear (LP) and quadratic programs (QP), consisting of a

---

[2]Consequently, the general E-ring formulation and code paths for exp and log terms are part of the framework design, but are not exercised in this preliminary empirical study.

mix between integer (ILP/IQP), mixed-integer (MILP/MIQP), and real-valued problems (LP/QP), which we have summarized in Table 3 in Appendix E.

Crucially, with the symbolic representation, we use standard regex methods to construct a default instance of Gurobi code, which we then execute to ensure that the problem has a pregenerated and annotated solution. Each generated problem is paired with a natural language representation. This is achieved by inserting the parameters of the formal optimization model into pre-built natural language templates. These templates are structured to mimic human-authored prompts, and include redundant sentence options to introduce variation to natural language representations, incorporating elliptical references,anaphora, ratio constructions, and domain-specific jargon. Variable and resource mentions are deliberately non-uniform to require coreference resolution rather than string matching. Appendix D.0.1 enumerates the pattern classes and provides verbatim examples for each, along with the sampling logic that yields lexical variety. Natural language prompts are assembled from templates using a combination of string formatting and regular expression substitution, ensuring that the resulting language is coherent and varied while remaining traceable to the underlying mathematical representation of an optimization problem. The initial batch of scenarios covers: Office supply budgeting, household budgeting, horticulture, cybersecurity staffing and management, military force structure, personnel management, and personal diet (such as macronutrient intake). Variables and resources are assigned labels from sets of available labels based on the selected context. Collectively, these statement sets are large enough to provide lexical diversity across problem statements exceeding those provided by Xiao et al. (2024).

We settled on the current hyperparameters so that roughly half the problems would be real-valued, 25% mixed integer, and 25% integer programs, three-quarters would be linear programs, and between 1/3 and 1/2 would be feasible. Achieving the latter required adjusting the hyperparameters used for generating the constraints. While we generate some instances of problems with hundreds to thousands of constraints, the majority of problems contain fewer than 36 constraints. We detail in Algorithm 1 the generation of a random instance of the Problem class, which we describe in detail in Appendix C.1.

## 3.2 Experimental Setup

We compare the NLMOptimizer dataset problems against the NL4OPT dataset (Ramamonjison et al., 2022a) given the latter's previous use as a benchmark against novel alternate datasets, its' public availability, its' suitability for zero-shot prompting, and for the number of problems available (contra the NLP4LP dataset consisting of 61 MILPs, NL4OPT has more than 600). We test two hypotheses: the first is that each LLM will perform worse on NLMOptimizer test problems than the NL4OPT benchmark dataset (e.g. lower end-to-end accuracy and higher errors across all experiments); the second is that expanded prompting to first reason out the specification of the optimization problem in the symbolic interchange format can improve the quality and success of the Gurobi code output from LLMs (that is, improved end-to-end accuracy and lower error rates). We test these two hypotheses using the following models: LLaMa-3.3 (Grattafiori et al., 2024), LLaMa-4-Scout-17B-16E-Instruct (AI, 2024), Gemini-1.5-Pro (Team et al., 2024), and GPT-OSS-120B (OpenAI et al., 2025). LLaMa-3.3 has a maximum number of tokens of 8000, whereas the other two models had a maximum number of tokens of 1000000. We tested each model and each problem set with two prompts: a base prompt that establishes the LLMService as a team of operations researchers and programmers tasked with producing Gurobi code to model and solve an optimization problem, and a second prompt expanding on the base prompt by providing additional instructions to conform their responses first to an IR format, before providing a code output. In both prompts, we feed the same natural language description of an OR problem.

We summarize the problems we generated from our Problem class for our experiment in Table 3, and in Figure 3 provide a cursory visualization of the differences in complexity between the NL4OPT dataset and the NLMOptimizer problems we have included in our supplementary materials. We note that we generated problems with substantially more variables than the NL4OPT problems, as well as those subject to multiple types of resources constraints. Whereas the NL4OPT dataset only contains problems constrained by one resource, such as budget, or space, our problems could have multiple different types of resources, leading to substantially more constraints while also producing realistic OR problems. We also explicitly cast constraints indicating variable type, as well as strictly asserting non-negativity. In our summary statis-

tics table, we denote all instances on our initial problem set by '-n'. We included the json and additional generation content from the NL4OPT project (Ramamonjison et al, 2022) in the dataset folder included in our supplementary materials. The NLMOptimizer problem set can be found in the 'oproblems' subfolder of the data folder included in our supplementary materials.

Following Xiao et al. (2024), we gathered end-to-end Accuracy (ACC), Compile Error (CE), and Runtime Error (RE), as well as adjusted end-to-end accuracy and runtime error scores. These suffice because the task inot a detection/-classification problem, and the outcome of interest is of constructive adequacy, which these scores directly capture. We note that for our purposes, recording end-to-end accuracy as is standard with these studies suffices ultimately due to the computational complexity of ensuring faithful representations of the optimization problems up to isomorphism, as many of the problems we generated had both redundant constraints or were otherwise infeasible, and our intende. End-to-end accuracy therefore serves as an acceptable proxy for true fidelity in problem representation for a preliminary investigation, while future work ought to expand

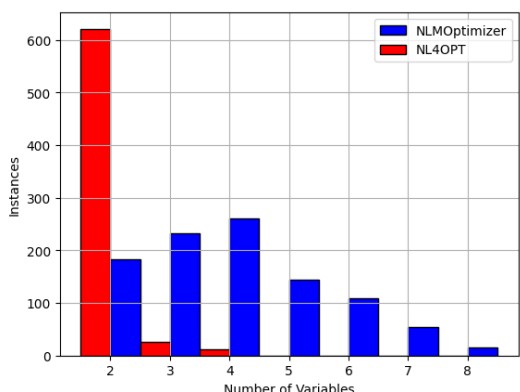

Figure 3: Complexity of the NL4OPT dataset versus our NLMOptimizer dataset with respect to number of variables in a problem instance.

to tolerate parsimonious, isomorphic representations of problems, while penalizing those adding extraneous, redundant constraints.

We implement Gurobi in Python 3.10.12 using Gurobi Optimizer version 12.0.2 build v12.0.2rc0 (linux64 - Ubuntu 22.04.4 LTS). All experiments were run using vanilla models, with the LLaMa models hosted locally. We set temperature to 0, top_p to 1, top_k to 0, and the frequency and presence penalty to 0. We implemented a 180 second timeout for each model, and when executing code via multiprocessing, implemented a 10 second timer, listing any code that exceeded this threshold as having a runtime error. All local components of the experiments were run with CPU model: AMD Ryzen Threadripper PRO 3995WX 64-Cores, instruction set [SSE2—AVX—AVX2], with a Thread count: 64 physical cores, 128 logical processors, using up to 32 threads. We include all prompts in Appendix D.

## 4 EXPERIMENTAL RESULTS

Table 1: Summary statistics by model, dataset, and additional prompting. Datasets are either NLMOptimizer (NLM) or the NL4OPT (nl4opt) datasets. Additional prompting denoted by sym.

| model&data&prompt. id | ACC | ACC-adj | CE | RE | RE-adj |
|---|---|---|---|---|---|
| llama3.3 NLM | 0.056 | 0.056 | 0.015 | 0.590 | 0.590 |
| llama4 NLM | 0.008 | 0.049 | 0.123 | 0.857 | 0.729 |
| gem NLM | 0.161 | 0.161 | 0.015 | 0.101 | 0.101 |
| gptoss NLM | 0.031 | 0.031 | 0.037 | 0.212 | 0.212 |
| llama3.3 NLM sym | 0.023 | 0.023 | 0.015 | 0.817 | 0.817 |
| llama4 NLM sym | 0.001 | 0.012 | 0.120 | 0.878 | 0.818 |
| gem NLM sym | 0.146 | 0.146 | 0.015 | 0.071 | 0.071 |
| gptoss NLM sym | 0.082 | 0.082 | 0.026 | 0.139 | 0.139 |
| llama3.3 nl4opt | 0.593 | 0.593 | 0.000 | 0.024 | 0.024 |
| llama 4 nl4opt | 0.017 | 0.035 | 0.003 | 0.995 | 0.894 |
| gem nl4opt | 0.660 | 0.660 | 0.000 | 0.027 | 0.027 |
| gptoss nl4opt | 0.065 | 0.065 | 0.000 | 0.005 | 0.005 |
| llama 3.3 nl4opt sym | 0.597 | 0.597 | 0.000 | 0.112 | 0.112 |
| llama 4 nl4opt sym | 0.061 | 0.064 | 0.005 | 0.960 | 0.954 |
| gem nl4opt sym | 0.666 | 0.666 | 0.000 | 0.020 | 0.020 |
| gptoss nl4opt sym | 0.184 | 0.184 | 0.000 | 0.000 | 0.000 |

We display the summary statistics of experiments by the model, dataset, and flag for additional prompting in in Table 1 (along with Figure 25 displayed in Appendix E) , and the Wilcoxon-Signed rank summary statistics in Table 2. Additionally, we plot the relevant statistics for each model and problem type across our different prompting scenarios in Figures 4 to 10, along with the breakdown of each model by experimental set-up. In these figures, each curve summarizes descriptive outcomes across the full 1000 instances provided in the dataset included in our supplementary materials. As these are aggregate outcomes over the provided dataset, they depict deterministic trends rather than sampling uncertainty, which could in principle be derived from gener-

ating multiple tranches with **Problem**. Finally, we provide expanded tables and figures in Appendix E, including a full audit of instance composition.

We make two immediate observations: across all four models, each performed appreciably worse with respect to end-to-end accuracy, compile error, and run-time errors, even after post hoc adjustment compared to the NL4OPT dataset, with poor performance observed across all problem types; independent of model, problem type, and prompt, end-to-end accuracy decline and errors increased for problems with more than 3 variables. Instances where model end-to-end accuracy improved tended to be spurious as models correctly concluded that the problems were infeasible, but otherwise failed to faithfully represent all named, albeit redundant and inconsistent constraints.

LLaMa-4-Scout-17B-16-Instruct was a particularly poorly performing model, in part because of near uniform failure to properly import Gurobi, which requires `import gurobipy`. Even when correcting for this, as noted by the adjusted end-to-end accuracy and adjusted RE scores, the run time errors remained high, and end-to-end accuracy remained below 10%. Improvements with respect to RE-adj did not lead to equivalent improvements in adjusted end-to-end accuracy, indicating that LLaMa-4 was especially challenged at both generating code and faithfully representing the optimization problem, independent of managing the proper imports.

Table 2: Wilcoxon signed rank one-sided hypothesis summary statistics (score, p-value)

|  | ACC | CE | RE |
|---|---|---|---|
| E1 (n=8) | (0,0.004) | (36,.004) | (29,0.074) |
| E2 (n=8) | (12,0.230) | (5,.250) | (18,0.527) |
| E2a (n=6632) | (61494,0.000) | (432,0.014) | (82956,1.000) |

In the case of additional prompting, the LLaMa-4 model managed to answer only 1 out of 1000 questions correctly without adjustment, after which it only managed to answer 12 questions correctly. LLaMa-3.3 also performed poorly on our problem set, but otherwise performed comparably to the Gemini model. Similar to LLaMa-4-Scout, GPT-OSS-120B performed poorly on both datasets but principally due to a failure to reliably capture problem description in the executable Gurobi code format that was requested. GPT-OSS-120B improved the most when including additional prompts to follow an IR template.

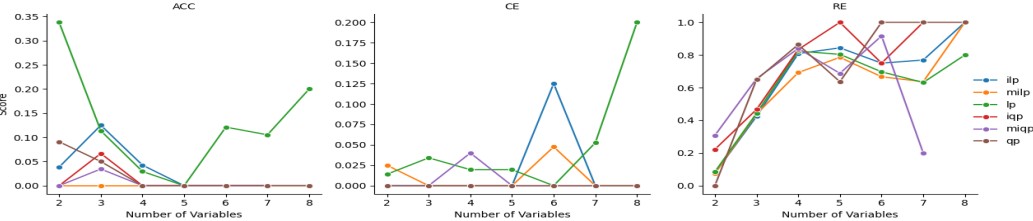

Figure 4: Distribution of scores for LLaMa-3.3 by number of variables.

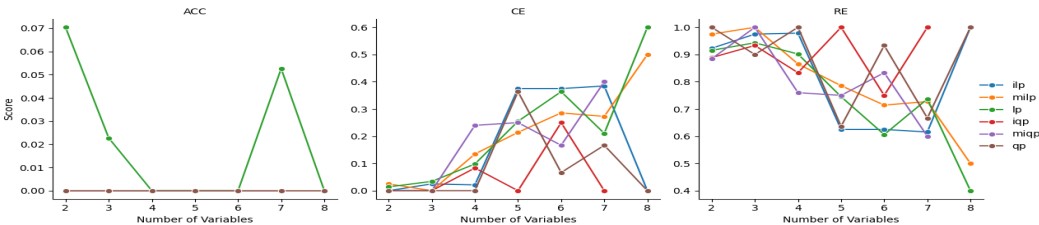

Figure 5: Distribution of scores for LLaMa-4 by number of variables.

The significance of these results for our two experiments, denoted by $E1$, $E2$ and $E2a$ are detailed in Table 2, where we gather the Wilcoxon statistic and corresponding p-value for changes to ACC, CE, and RE results across all experiments. Further, while we can only compare the point-estimates

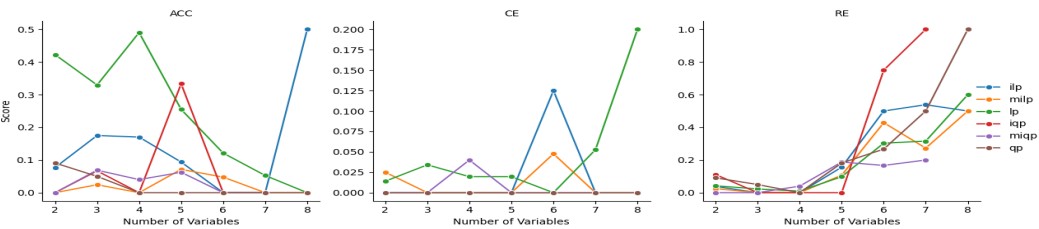

Figure 6: Distribution of scores for Gemini by number of variables.

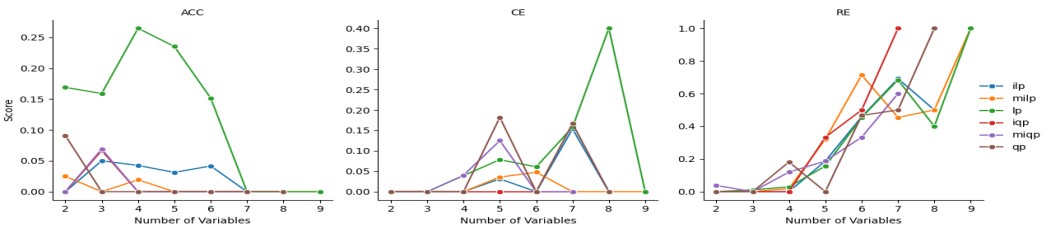

Figure 7: Distribution of scores for GPT-OSS with additional prompting by number of variables.

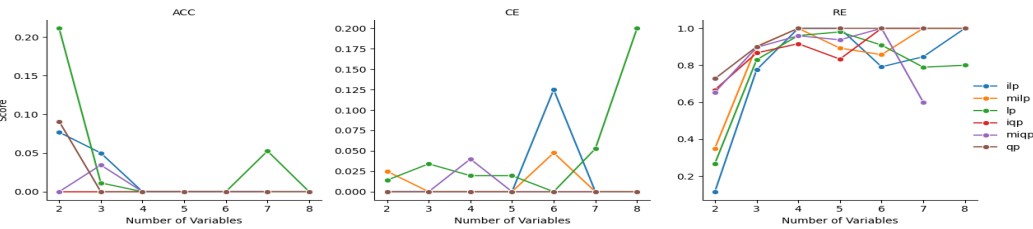

Figure 8: Distribution of scores for LLaMa-3.3 with additional prompting by number of variables.

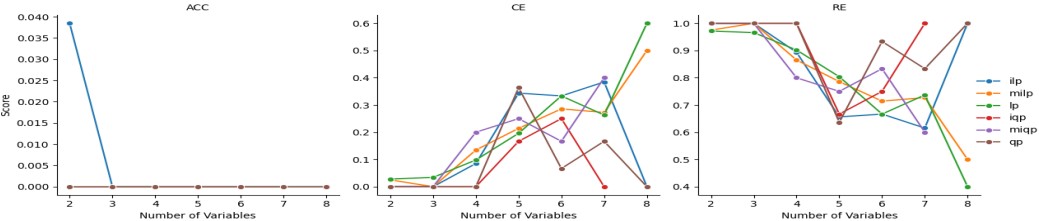

Figure 9: Distribution of scores for LLaMa-4 with additional prompting by number of variables.

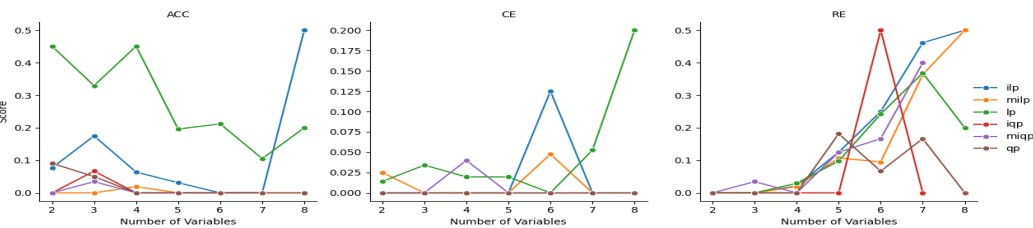

Figure 10: Distribution of scores for Gemini with additional prompting by number of variables.

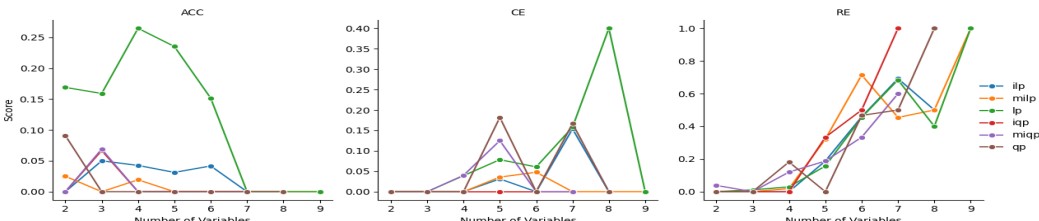

Figure 11: Distribution of scores for GPT-OSS with additional prompting by number of variables.

between models for experiment 1, we can also look at the difference between prompts on a problem-by-problem basis in experiment 2, with the problem-by-problem results displayed as $E2a$, and have developed a protocol (see Figure 2) for future dynamic evaluation. We list the sample sizes respectively for the test. In $E1$ we match against the two datasets, in $E2$ we match against the results of using the same base prompt for each model against an expanded prompt first specifying a preferred symbolic structure. For the first experiment: we reject the null-hypothesis against the alternative hypothesis that benchmarking NL4OPT against the original problems leads to an average accuracy difference below 0 (i.e. models will be less accurate with respect to the novel data presented by the Problem class), and we reject the null-hypothesis against the alternative hypothesis that benchmarking NL4OPT against the original problems leads to an average Compilation Error difference greater than 0 (i.e. models will have more compilation errors when trying to model the original problems in the Problem class); we fail to reject the null-hypothesis for run-time errors. For the second experiment, both with respect to the point-estimate as well as the problem-by-problem basis, we fail to reject the null-hypothesis against the alternative hypotheses where the additional symbolic prompting improves accuracy (or decreases compile time and run time errors) over the baseline prompting if we group by model. However, when comparing problems directly, we found statistically significant improvements with respect to accuracy and compilation errors when including additional prompting.

## 5 CONCLUSION

Our findings are consistent with the observations in Xiao et al. (2024) and Mostajabdaveh et al. (2025): even the best models plateau well below expert performance on modeling-centric questions, finding further evidence that general-purpose instruction-tuned LLMs, even with explicit chain-of-thought or chain-of-experts implementations, fail on more realistic LP/QP instances across those respective ablation studies, and that errors concentrate in categories requiring identification of entities and relations, reinforcing our finding that the core bottleneck is problem representation. We conclude that the NL4OPT competition dataset is less appropriate as a training and benchmarking dataset than problems generated by our Problem class for the task of generating code that correctly solves an optimization problem. The problems in NL4OPT are too simple to be representative of actual optimization problems faced by real-world operations researchers. We further find support that the alternate benchmark datasets discussed in this paper are inadequate for the task of *representing* OR problems for training. We are not proposing a faster solver, nor that MCQ should be abandoned as a complementary benchmark for understanding. We advocate for the adoption of the **Problem** generator class over other alternatives for future model development on the grounds that ours covers a substantially larger class of problems that are relevant to both the real-world OR community, and that training, testing and validation can be done dynamically using the NLMOptimizer protocol so as to avoid overfitting a static dataset. Finally, while outside the scope of our experiments, we also explored initial human-guided interaction between LLMs using a Chain-of-Expert agents approach coupled with the **SymInterchange** class methods and real-time display of problem formulation. We found this successfully improved conversion of problems from natural language to executable Python code that also properly solved the problem for LLaMa-3.3 and Gemini-1.5-Pro. We encourage future research in the direction of multi-shot prompting LLM-agents to use both the Problem class and the corrective methods for the SymInterchange class.

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

## A    APPENDIX

A full reproduction of the Problem class will not fit in an Appendix, but has been included in the Supplementary materials. However, we have provided details for the hyperparameters we used for generating the 1000 problems used for benchmarking, the experimental set-up, the natural language problem generation, and some additional breakdown of the performance of the LLMs.

## B    THEORETICAL BACKGROUND

Our choice of E-rings as the principle representation for OR optimization problems relies on a pragmatic argument that this is a suitable formal object for representing a wide class of problems of interest that are largely decidable, and such that intermediate representations of definable sets are amenable to neurosymbolic to determine if problems are solvable.

Throughout this Appendix, we expand on the material in Section 2.2, providing the readers primers on quantifier elimination (QE), o-minimality, and E-rings, along with corresponding results that provide decidability guarantees.

### B.1 OPTIMIZATION FORMALLY UNDERSTOOD

We first summarize the primary scope of optimization problems considered in the main body of the paper.

*Linear programs* involve a linear objective function over a polyhedron defined by linear equalities and inequalities, whereas *quadratic programs* allow for the objective function to be a quadratic function. Convex optimization occupies a central place in this landscape, characterized by problems where both the objective function and feasible region are convex (Jensen & Bard, 2002). Convexity guarantees the existence of global minima and underpins the success of scalable first- and second-order methods, however it is not enough to reveal the algebraic structure of optimization problems or to build a symbolic framework for constraint representation for the full scope of optimization problems that concern us (Aubin, 2002). For LLM systems that generate optimization routines, a symbolic formalism capturing the full spectrum of objective functions and constraint types is essential for expressivity and correctness.

### B.2 LANGUAGES AND DEFINABILITY

Following Marker (2002), we recall the definition of languages, structure, and definability.

**Definition B.1.** *A first-order language $\mathcal{L}$ is given by specifying the following data:*

1. *a set of function symbols $\mathcal{F}$ and positive integers $n_f$ for each $f \in \mathcal{F}$ describing the arity of the function;*

2. *a set of relation symbols $\mathcal{R}$ and positive integers $n_R$ for each $R \in \mathcal{R}$;*

3. *a set of constant symbols $\mathcal{C}$*

*An $\mathcal{L}$-structure $\mathcal{M}$ is a given by the following data:*

1. *a nonempty set $M$ called the universe, domain, or underlying set of $\mathcal{M}$;*

2. *a function $f^{\mathcal{M}} : M^{n_f} \to M$ for each $f \in \mathcal{F}$;*

3. *a set $R^{\mathcal{M}} \subset M^{n_R}$ for each $R \in \mathcal{R}$;*

4. *an element $c^{\mathcal{M}} \in M$ for each $c \in \mathcal{C}$.*

*$f^{\mathcal{M}}, R^{\mathcal{M}}$ and $c^{\mathcal{M}}$ are refered to as the interpretations of their respective symbols. The set of $\mathcal{L}$-terms is the smallest set $\mathcal{T}$ such that*

1. *$c \in \mathcal{T}$ for each $c \in \mathcal{C}$;*

2. *each variable symbol $v_i \in \mathcal{T}$ for i=1,2,3,…;*

3. *if $t_1, \ldots, t_{n_f} \in \mathcal{T}$ and $f \in \mathcal{F}$, then $f(t_1, \ldots, t_{n_f}) \in \mathcal{T}$.*

*We say $\phi$ is an atomic-formula if $\phi$ is either*

1. *$t_1 = t_2$ for terms $t_1, t_2$;*

2. *$R(t_1, \ldots, t_{n_R})$ for terms $t_1, \ldots, t_{n_R}$.*

*The set of $\mathcal{L}$-formulas is the smallest set $\mathcal{W}$ containing the atomic formulas and closed under logical connectives and quantifiers, i.e.*

1. *if $\phi \in \mathcal{W}$, then $\neg\phi \in \mathcal{W}$*

2. *if $\phi, \psi \in \mathcal{W}$, then $\phi \wedge \psi$ and $\phi \vee \psi$ are in $\mathcal{W}$;*

3. *if $\phi in \mathcal{W}$, then $\forall v_i \phi \in \mathcal{W}$ and $\exists v_i \phi \in \mathcal{W}$.*

*Given an $\mathcal{L}$-structure $\mathcal{M}$, a subset $X \subseteq M^n$ is* definable *if $X = \{x \in M^n : \varphi(x)\}$ for some first-order $\mathcal{L}$-formula $\varphi$ with parameters from $M$.*

Let $\mathcal{L}_{\mathrm{or}} := \{0, 1, +, \cdot, \leq\}$ be the language of ordered rings. Let $\mathcal{L}_{\exp} := \mathcal{L}_{\mathrm{or}} \cup \{\exp\}$ be the language of ordered exponential rings (*E-rings* in the sense used in this paper). We write $\mathbb{R}_{\mathrm{or}}$ for the real field in $\mathcal{L}_{\mathrm{or}}$ and $\mathbb{R}_{\exp}$ for $(\mathbb{R}; 0, 1, +, \cdot, \leq, \exp)$. For restricted analytic functions, write $\mathbb{R}_{\mathrm{an}}$, and for the expansion by the (unrestricted) exponential function, $\mathbb{R}_{\mathrm{an},\exp}$.

Further, we recall the chain of definitions for real-closed fields and semi-algebraic sets:

**Definition B.2.** *For any of the languages $\mathcal{L}$ whose underlying universe is a field $F$, we say $F$ is real-closed if there is a total order $<$ on $F$ such that every positive element of $F$ has a square root in $F$ and every polynomial of odd degree with coefficients in $F$ has at least one root in $F$. The canonical real-closed field is the field of real numbers, $\mathbb{R}$. For any real-closed field $F$, a set $\mathcal{W} \subset F^n$ is said to be basic closed semialgebraic set if it is a finite intersection of sets defined by polynomial inequalities of the form $p_i(X_1, \ldots, X_n) \geq 0$ for $p_i \in F[X_1, \ldots, X_n]$, so that $\mathcal{W} = \{\vec{x} \in F^n \mid \bigwedge_{i \in I_W}^r p_i(\vec{x}) \geq 0\}$, while a general semialgebraic set is a finite Boolean combination of basic closed algebraic sets (Netzer, 2016; Marker, 2002).*

With respect to our semi-algebraic set, this means $\varphi(a)$ satisfies the implied (in)equalities expressed in formula $\varphi$.

In theory, optimization problems rely on feasible sets, and in practice feasible sets and the objective functions are often precisely those that are definable in terms from the first-order language of ordered commutative rings $\mathcal{L}_{or} = \langle 0, 1+, \cdot, < \rangle$. In our expansion of the language $\mathcal{L}_{or}$ to include a function symbol $E$, or $\exp$, we want to capture semi-algebraic sets that definable in logarithmic-exponential polynomial terms, and provide sentences that allow us to interpret $E$ as the standard real-valued exponential function. We will summarize below the model theoretic reasoning as to why this is both desirable for optimization by SDP, as well as how o-minimal geometry allows us to retain the decidability necessary to determine feasibility.

As mentioned before, it is immediate that there is a bijective correspondence between the polynomials in any given ordered ring $R[X_1, \ldots, X_m]$ with indeterminates $X_i$, and the terms generated from $\mathcal{L}_{or}$ when adding the constants of $R$ to the $\mathcal{L}_{or}$ (without any extensions of the language, the terms are precisely in correspondence with polynomials in $\mathbb{Z}[X_1, \ldots, X_m]$) (Marker, 2002).

Expanding on E-rings in greater detail and following (Van Den Dries, 1984), an *E-ring* is a pair $(R, E)$ where $R$ is a ring with unity, denoted by 1, and a map $E : (R, +) \to R_u^\times$, such that $E(0) = 1$, and which maps the additive group structure of $R$ to the multiplicative group of units of $R$, so that $E(x+y) = E(x) \cdot E(y)$. Given an E-ring (R,E), the ring of *E-polynomial* in indeterminates $X_1, \ldots, X_m$ over $R$, is denoted by $R[X_1, \ldots, X_m]^E$, and has the structure of a group ring over the polynomial ring $R[X_1, \ldots, X_m]$. The additive group structure of the E-polynomial ring is given by: $R \oplus \bigoplus_{k=0}^{\infty} A_k$, where $A_k$ is recursively defined in terms of $A_k$, group homomorphisms $E_k$, and rings $R_k$ such that $R_k \subset R_{k+1}$ and $E_{k+1}$ is a functional extension of $E_k$, which is a group homomorphism from the additive group $R_k$ to the group of units in $R_{k+1}$. In particular, each $E_k : R_k \to R_{k+1}$ is a group-homomorphism sending each $r \in R_k$ written as $r = r' + a$ such that $r' \in R_{k-1}$ and $a \in A_k$ to the element $E_{k-1}(r') \exp(a)$ in $R_{k+1}$, such that $R_{k+1} := R_k[\exp(A_k)]$, the group ring of $\exp(A_k)$ over $R_k$. In turn, $A_k$ is defined to be an $R_k$-submodule of $R_{k+1}$ freely generated by $\exp(a)$ with $a \in A_k$ for $a \neq 0$, which establishes $R_{k+1} = R_k \oplus A_{k+1}$ as additive groups. In turn, the underlying polynomial ring $R[X_1, \ldots, X_m]^E$ is taken as the algebraic object $\varinjlim R_k = \bigcup_{k \in \mathbb{N}} R_k$.

Of particular interest, any polynomial $p \in R[X_1 \ldots, X_m]^E$ will correspond to a formal term $t_p \in R \oplus \bigoplus_{k=0}^{\infty} A_k$, which in turn is an element $t_p \in R_k$, where $k$ is of *height* of $t_p$, with $k$ the integer where $t_p \in R_k \backslash R_{k-1}$. Intuitively, the height is the maximum number of embedded exponentiations appearing in the term $t_p$. Real exponential rings introduce a compositional algebraic structure capable of representing optimization constraints involving both polynomials and exponentials in a uniform symbolic language. This algebraic expressiveness invites a deeper inquiry into the geometric behavior of such constraints.

### B.3 QUANTIFIER ELIMINATION AND DECIDABILITY IN REAL CLOSED FIELDS

We now describe two foundational theorems for our purposes: Theorems B.3 and B.4, (see Seidenberg (1954) and Collins (1976) for full proofs and details).

**Theorem B.3** (Tarski-Seidenberg). *The complete first-order theory* $\mathrm{Th}(\mathbb{R}_{\mathrm{or}})$ *of real closed ordered fields admits* quantifier elimination. *In particular, any first-order sentence over* $(\mathbb{R}; 0, 1, +, \cdot, \leq)$ *is effectively reducible to a quantifier-free one, hence* $\mathrm{Th}(\mathbb{R}_{\mathrm{or}})$ *is decidable.*

**Theorem B.4.** *There is an effective procedure,* cylindrical algebraic decomposition *(CAD), that performs quantifier elimination over* $\mathbb{R}_{\mathrm{or}}$. *The worst-case complexity is doubly-exponential in the number of variables, and this is optimal in general.*

Theorem B.3 implies that the image of any semi-algebraic set under a polynomial map remains semi-algebraic under coordinate projection, which is the geometric core of quantifier elimination over the reals (Seidenberg, 1954). Concretely, existential quantifiers, and thus universal quantifiers under the identification that $\forall x \varphi(x) \equiv \neg \exists x \neg \varphi(x)$, over real variables can be eliminated effectively, reducing feasibility questions for semi-algebraic constraints to equivalent quantifier-free formulas that symbolic procedures can manipulate. Algorithmically, Collins' cylindrical algebraic decomposition (CAD) provides a constructive QE method: it decomposes $\mathbb{R}^n$ into finitely many *cells* on which every input polynomial has a constant sign, enabling projection and elimination steps that decide first-order sentences in the language of ordered rings. (Collins, 1976) While CAD is worst-case doubly exponential in the number of variables, and this is unavoidable in general, its cell or sample-point outputs yield exact certificates of feasibility and optimality for low-dimensional or low-degree instances. Further, guarantees of decidability provide motivation for reliance on approximation methods which can be run more efficiently and for real-world applications often suffice given tolerable imprecision found within real-world implementations.

In addition to decidable guarantees for feasibility, real algebraic geometry is furnished with many decision-to-optimization reductions and critical-point methods that recover algebraic minimizers by eliminating variables (and multipliers) from KKT-style conditions or by optimizing on QE-produced cells (Basu et al., 2006). These facts are what justify using semi-algebraic structure as a target for our IR and validators: feasibility remains a decidable, tame geometric problem; optimality can be certified exactly for small cases; and larger cases still benefit from the same logical invariants (e.g., unit/convexity checks) even when solved numerically(Basu et al., 2006). We summarize these observations for practical purposes in Corollary B.5.

**Corollary B.5** (Decision-to-optimization reduction). *Let* $f, g_i, h_j$ *be polynomial terms, and define a decision predicate* $P(t) := (\exists x) \bigwedge_i g_i(x) \leq 0 \wedge \bigwedge_j h_j(x) = 0 \wedge f(x) \leq t$. *Then* $P$ *is decidable.*

The proof of Corollary B.5 follows immediately by Theorem B.3. As a consequence of Corollary B.5, the global minimum $\min\{f(x) : g_i(x) \leq 0, \ h_j(x) = 0\}$ is computable exactly by bracketing $t$ and deciding $P(t)$, or by eliminating $x$ from the KKT/global optimality conditions.

**Remark B.6** (What QE means in practice). *For low degrees, i.e. small $n$, CAD-based QE supplies exact feasibility certificates, algebraic optima, and sample points for minimizers. For larger degrees, best practices rely on numerical solvers. Nonetheless, these problems still retain the logical* form *to guide validation.*

### B.4 O-MINIMAL EXPANSIONS AND EXPONENTIALS

Let $\mathcal{L}$ contain a relation symbol $<$, and let $\mathcal{M}$ be a dense linear order with respect to $<$. Then $\mathcal{M}$ is *o-minimal* iff every definable subset $S \subset M$ (with parameters) is a finite union of points and intervals. Following Van den Dries (1998), o-minimality generalizes the semialgebraic setting once $\mathcal{L}$ furnishes an ordered ring structure, so that *tame* sets behave like semialgebraic sets with robust geometric control (monotonicity, cell decomposition, dimension). Practically, the connective tissue to optimization is that tame geometry underlies semidefinite *relaxations* for polynomial optimization, notably Lasserre's hierarchy: minimizing a polynomial $f$ on a compact basic semialgebraic set $\mathcal{W}$ is approached by a sequence of SDPs over moment cones with SOS certificates (Josz & Molzahn, 2018). Each level optimizes a linear functional subject to PSD moment/localizing matrices, encoding polynomial nonnegativity by SOS constraints and yielding increasingly tight lower bounds, with many refinements for sparsity and large scale. Conceptually, this casts global polynomial optimiza-

tion over semialgebraic sets as an SDP sequence that exploits positivity, duality, and moment/SOS geometry(Netzer, 2016; Josz & Molzahn, 2018).

**Theorem B.7** (O-minimality of $\mathbb{R}_{an}$). *The structure $\mathbb{R}_{an}$ (real field expanded by all restricted analytic functions) is o-minimal. Consequently, definable sets admit cell decomposition and the usual tame geometric properties. (van den Dries et al., 1994)*

O-minimality yields a calculus for tame sets and functions: finite cell decompositions, stratifications, and dimension control that mirror the semialgebraic case and support symbolic preprocessing (projection, variable elimination) in optimization pipelines (Van den Dries, 1998). As explored in Attouch et al. (2013), tameness also underpins algorithmic analysis, as for tame (e.g., semialgebraic/definable) objectives, descent and proximal methods will still admit global convergence guarantees via the Kurdyka–Łojasiewicz inequality. In particular, definability ensures the KŁproperty and yields convergence rates and stability for a wide class of first-order schemes.

Real-world models often require exponentials (entropy, logistic regression, log-barriers, softmax/-soft constraints), and we want to retain tameness while adding $\exp$ (Wilkie, 1998). The key bridge we identify builds upon Wilkie's results for exponential fields (principally Theorems B.8 and B.9): expansions of the real field by suitable analytic functions, including the (unrestricted) exponential, remain o-minimal or at least admit strong model-theoretic control, and so definable sets with exponentials inherit the same geometric regularity (cell decomposition, dimension, finiteness properties) (Wilkie, 1996). This provides the fundamental justification for our choice of *E-rings* as the target for our IR. We can model exponential terms while preserving tame geometry for symbolic reasoning and quantifier simplification (van den Dries et al., 1994).

**Theorem B.8** (Wilkie's Theorem). *The real exponential field $\mathbb{R}_{exp}$ is o-minimal, and certain expansions by restricted Pfaffian functions are model complete. (Wilkie, 1996)*

Following Wilkie (1999), o-minimality of $\mathbb{R}_{exp}$ gives tame geometry for formulas involving real polynomials and exponentials: definable sets admit cell decomposition and controlled combinatorics, which is precisely the regularity needed to design and analyze robust optimization layers (e.g., screening infeasibility, preserving convexity, bounding active set changes). This directly supports our intended natural language to IR workflow that must manipulate exponential constraints without leaving the tame world.

**Theorem B.9.** *(van den Dries et al., 1994) The expansion $\mathbb{R}_{an,exp}$ is model complete and o-minimal.*

Theorems B.7 and B.9 are forms of a *weak* QE result. Whereas *strong* QE means every first-order formula in the given language is equivalent to a quantifier-free one in the *same* language, as in real closed fields, by contrast, there are two practically important senses of *weak* QE that arise in o-minimal expansions such as $\mathbb{R}_{an,exp}$. First, the *model-completeness* sense, where every formula is equivalent to an *existential* formula (no universal blocks or alternation) though not necessarily quantifier-free. Second, the *language-expansion* sense, where formulas become quantifier-free after enlarging the language by naming standard definable primitives (e.g., restricted analytic/Pfaffian pieces, $\exp$), even if they are not quantifier-free in the strict base language (Van Den Dries, 1984; Wilkie, 1996; Van den Dries & Miller, 1996; Marker, 2002).

Either sense of weak QE enables a solver architecture resembling those employed by mixed-integer linear and cone programs with cutting planes and oracles, as the solution to an optimization problem is reduced to a $\Sigma_1$- sentence of the form $\exists \vec{x}[f(\vec{x}) \leq t \wedge \varphi(\vec{x})]$, where $f$ is a (definable) objective function, $\varphi$ is the collected formula describing the constraints, and $t$ is the definable candidate term for an optimal solution (in this particular setting, $f$ and $t$ are scalar valued, but this can be further amended for vector-valued functions, and general definable functions). In particular, existential checks can be approached with feasibility oracles, and the optimization loop then is recast as repeated queries checking whether a current discrete assignment or candidate bound term for the objective function returns a *witness* to the defining formula for the objective modulo constraints (a sample point with the corresponding algebraic data), or a small unsatisfiable explanation, from which we can then deploy branch-and-bound and cut-generation loops as needed. Future work involving formal verification can post-certify winners via Positivstellensatz or KKT-based checks (see Netzer (2016) for details), following common OR workflows validating incumbent solutions instead of proving global optimality at every node. Further, in the weak QE setting, the polynomial term constraints can be cast into an answer-set program (ASP), where they act as external atoms that are evaluated by an oracle given the current ASP assignment to discrete choices and parameters symbols.

For our pipeline, this distinction guides how we validate and compile $NL \to IR$ instances. In the *semialgebraic* fragment, strong QE yields full elimination and decidability, which we exploit for exact feasibility/optimality on small cases and for sound structural checks more generally. In the *exponential* fragment, $\mathbb{R}_{an,exp}$ gives o-minimal tameness and model-completeness (weak QE). In this general setting, feasibility reduces to existential forms, projections preserve definability, and quantifiers can be simplified once the IR names the right primitives. O-minimality/model completeness do *not* by themselves yield decidability. The decidability of $\mathrm{Th}(\mathbb{R}_{exp})$ is *open*. Assuming a suitable real form of Schanuel's conjecture, Macintyre and Wilkie show that $\mathrm{Th}(\mathbb{R}_{exp})$ is decidable; conversely, a *weak Schanuel* statement is equivalent to decidability(Wilkie, 1997).

## B.5 CONSEQUENCES FOR OR MODELING

Reiterating our motivation, many optimization models used in OR and ML are definable in the languages described above expressing tame theories: LP, QP, Second Order Cone Programs (SOCP), and SDP feasible regions are semialgebraic, and common exponential terms (such as entropy, log-likelihoods, log-barriers, softmax) live naturally in expansions with $\exp$. Working in an o-minimal expansion preserves geometric regularity (chiefly cell decomposition, dimension theory, stratifications), which in turn supports symbolic preprocessing (namely projection and thus elimination) and robust certification prior to numerical solution (Denef & Van den Dries, 1988; Van den Dries & Miller, 1996). For polynomial parts, Lasserre-type hierarchies provide systematic SDP relaxations and certificates via moments/SOS, allowing us to transform global polynomial optimization over compact semialgebraic sets into a convergent sequence of tractable SDPs (Josz & Molzahn, 2018; Netzer, 2016).

**Proposition B.10** (Standard OR cones are definable). *The feasible sets of LPs and QPs are semialgebraic (linear or quadratic equalities/ inequalities). SOCP constraints $\|Ax + b\|_2 \le c^\top x + d$ are semialgebraic. SDP constraints $\{X \succeq 0\}$ are semialgebraic via principal minors. Exponential/power cones (e.g., $y \exp(x/y) \le z, \ y > 0$) are definable in $\mathbb{R}_{exp}$ and hence in $\mathbb{R}_{an,exp}$.*

Prop. B.10 justifies aiming the natural language to IR mapping at a formal language whose first order terms are composed of polynomials and $\exp$: we keep the cones that appear in OR, while staying within an o-minimal setting. This yields closure under projection and cell decomposition—key for IR-level checks (unit consistency, convexity screens, feasibility typing) and safe simplifications before calling a solver. For purely polynomial instances, moment/SOS machinery supplies SDP-based lower bounds and, in small cases, exact certificates.

Concretely, the E-ring perspective provides a first-order language in which constraints take the form

$$\varphi(\mathbf{x}) \equiv t_p(\mathbf{x}) \ge 0,$$

where $t_p$ is a formal term built from polynomials and (possibly) $\exp$. Then, via the established correspondence, $t_p$ is in one-to-one correspondence with a polynomial in an E-ring understood as nested compositions of elements of $\mathbb{R}[X_1, \ldots, X_m]$ and $\exp$. In this language, *feasibility* corresponds to the existential truth of a sentence, while *optimality* (e.g., minimality) requires restricted universal quantification. O-minimality provides the weak quantifier-elimination and geometric tameness needed so that such transformations remain within the definable universe (Wilkie, 1996; van den Dries et al., 1994; Marker, 2002). We reiterate and gather this observation in the following Corollary:

**Corollary B.11** (Pragmatic E-ring policy). *If an $NL \to IR$ mapping stays within $\mathbb{R}_{or}$ ( namely for LP/QP/SOCP/SDP), feasibility/optimality are* decidable *and admit effective QE. If an $NL \to IR$ mapping stays within $\mathbb{R}_{an,exp}$, we admit weak quantifier elimination.*

Corollary B.11 follows principally from Theorems B.3 – B.4. When exponentials appear in terms, we will be working in $\mathbb{R}_{an,exp}$, which by Theorems B.8–B.9 yields tame geometry and model completeness. Although full decidability is not known in general, but the o-minimal setting supports robust validation and quantifier *simplification*, as for fragments that appear within $\mathcal{L}_{or}$, we will have decidability guarantees. This justifies our choice to target the E-ring fragment for modeling and training.

We note that for terms in these languages With integer variables, the linear fragment (Presburger Arithmetic, i.e. an ILP) is decidable. If we allow for general polynomial relations over $\mathbb{Z}$ though, this leads to undecidability as determined by Hilbert's Tenth Problem. The proposed IR conforming to **Problem** therefore classifies integer problems and restricts to the linear case for exact certification.

Finally, given a **Problem** instance in the semi-algebraic fragment, we may form $P(t)$ as in Cor. B.5. Using CAD (or a specialized QE) to decide $P(t)$ along a rational bisection on $t$, we can then extract $t^\star$ and (optionally) a sample minimizer via the CAD cell containing the optimum. For larger instances, we may still use numerical solvers but keep IR-level feasibility/convexity/unit checks as first-order invariants. For the purposes of improving LLM-based systems then, reliably reconstructing and building an IR that maps faithfully onto a **Problem** instance is a sufficient goal for operations research optimization from natural language.

# C  CODE

## C.1  GENERATING A PROBLEM

We examine in greater detail the pseudo-code description provided in the main body of the paper, so that readers can examine the code we have provided in the supplementary materials with greater clarity.

Although we provide methods to pre-load problem configurations, by default we randomly generate problems from scratch with a number of fixed hyperparameters, both in the class __init__ method, as well as within various functions. $n_{vars}$, the number of variables, and $n_{res}$, the number of resources are generated from a mixed model of three separate integral uniform distribution and hypergeometric distributions respectively, each determined by three hyperparameters. Because minimization problems and maximization problems can be freely converted between each other by multiplying the objective function by $-1$, we used a fair Bernoulli random variable to determine the problem goal.

We only generate arbitrary polynomials in the base E-ring in order to compare the complexity of our linear and quadratic programs with the linear programs found in the NL4OPT dataset. The objective function is generated by the _gen_function method of the problem class, and randomly selects monomials by the grading of the degree from combinations of the variables. Coefficients for each monomial term are either drawn uniformly from $Unif_{\mathbb{Z}}(1, 10)$ or $Unif_{\mathbb{Z}}(1, 10) + Unif(0, 1)$ depending on an fixed hyperparameter for whether we prefer integer coefficients only. We have opted to default solely to integer coefficients.

Constraints are organized into four cases: non-negativity, type, lower-bound, and upper-bound. The non-negativity constraints were implemented to ensure that all variables are to be non-negative. The type constraints are used to enforce cases where variables must be integral - if any variable is integral, the problem becomes mixed-integer unless all variables are integral, in which case it becomes an integer program. Lower- and upper-bound constraints were further sorted into several resources that are randomly determined by a class hyperparameter.

Lower- and upper-bound constraints consist of either two, three or all variables, and either determine a minimum or maximum proportion for those constraints or otherwise correspond to a minimum or maximum allocation of resources set by a budget cap that is by default randomly generated along with the scalar coefficients for each resource. The coefficients appearing in both budget constraint types are randomly generated by default with the _gen_resources_dict method unless users provide explicit resource functions when initializing a Problem instance. The default method first generates three random integers: $a$ and $b$ are uniformly selected from between 2 and 10, while $t$ is drawn $t \sim 2 + HGeo(a, b, n), n \sim Unif(a, a + b)$. Afterward, with a fixed hyperparameter value of resources_split_parameter, which we default to .2, if a variable is a uniform random variable from the unit interval is greater than this hyperparameter, then the coefficients were drawn from a random integer matrix whose components were uniformly drawn between 1 and 3t; otherwise, coefficients were drawn from a real-valued random matrix whose components were drawn uniformly between 1 and 3t and rounded to the hundredth's place. Similarly, budget cap 'seeds' $b_{res}$ are used when generating constraints as a random vector drawn uniformly from between L and U, where $L \sim Unif_{\mathbb{Z}}(2tn_{vars}, 5tn_{vars})$ and $U \sim Unif_{\mathbb{Z}}(6tn_{vars}, 20tn_{vars})$. For the resource constraints, bounds are then uniformly drawn from $Unif_{\mathbb{Z}}(\lfloor b_{res}/(3n_{vars})\rfloor, \lfloor b_{res}/(n_{vars})\rfloor)$ for lower bounds, and $Unif_{\mathbb{Z}}(\lfloor b_{res}/(n_{vars})\rfloor, b_{res})$ for upper bounds.

The number of pairs and triples that appear in the lower- and upper-bound constraints for each resource type are random variables, which depend on whether the goal is to maximize or minimize

the objective function. We describe the minimize case below, noting that we reverse the parameters for the lower and upper bounds when the goal is to maximize the objective function. These variables are $n_{lpair,res}, n_{ltriple,res}, n_{upair,res}, n_{utriple,res}$ and in the minimization case are drawn as follows:

$$r \sim Unif_{\mathbb{Z}}(n_{vars}, \binom{n_{vars}}{2})) \qquad s \sim Unif_{\mathbb{Z}}(0, \binom{n_{vars}}{3}))$$

$$t \sim Unif_{\mathbb{Z}}(0, \binom{n_{vars}}{2})) \qquad u \sim Unif_{\mathbb{Z}}(0, \binom{n_{vars}}{3}))$$

$$n_{lpair,res} \sim HGeo(r, \binom{n_{vars}}{2}), \lfloor \frac{3r}{2} \rfloor)$$

$$n_{ltriple,res} \sim HGeo(s, \binom{n_{vars}}{3}), \lfloor \frac{3s}{2} \rfloor$$

$$n_{upair,res} \sim HGeo(t, \binom{n_{vars}}{2}), \lfloor \frac{3t}{2} \rfloor)$$

$$n_{utriple,res} \sim HGeo(u, \binom{n_{vars}}{2}), \lfloor \frac{3u}{2} \rfloor)$$

Having now formed the symbolic representation of the problem, we now randomly select the semantic template for the natural language description of the problem. The options are conditioned on the `problem_type` attribute, and are drawn uniformly from one of three possible lists. Then, depending on the number of variables in the symbolic problem, we uniformly select from the semantic template determined by the `semantic_problem_type` attribute, we select corresponding variables from a list of possible semantic variable names with the `_gen_sym_vars` method. Similarly, we select resource names in a similar fashion with the `_gen_nl_resources_dict` method. After this, we construct the word problem through substitution and random selection of syntactically correct connecting phrases. We explain this in greater detail in the Data section of this Appendix.

We also convert the symbolic description of the problem into a corresponding Python file that implements the problem in Gurobi. We default to Gurobi in this paper, as we have restricted ourselves for the time being to quadratic problems. However, when dealing with arbitrary terms drawn from an E-ring, we would default to an implementation in SymPy. This will require a future iteration of the Problem class to substitute the `gurobi_code` attribute with a generic `code` attribute name. With respect to the translation of the problem into Gurobi code, this amounts to properly providing the Problems symbolic attributes for the objective function, variables, and constraints as they appear in attribute dictionaries. Finally, we run the code produced with the `_run_gurobi_code` method, storing it as the `problem_solution` attribute.

## C.2 RUNNING EXPERIMENTS

We have included a detailed README file in our supplementary materials. We advise researchers consult this first before running any experiments. We also advise researchers to properly provide their own API keys for each respective model, or appropriately adjusting the LLM Service templates that we have provided in order to ensure smooth operation. We include two functions to run experiments on the NL4OPT data and the NLMOptimizer problems with the baseline_llm_nlp4opt_qa.py and baseline_llm_problem_answer.py files respectively. These functions run locally within the supplementary materials folder and interact with the Data and Dataset folders also included in the Supplementary materials folder.

For the NL4OPT data, we convert the stored problem description into our Problem class format, primarily by preloading in the symbolic representation of the problem and natural language description of the problem, before also generating appropriate Gurobi code. All experiments are run through an interface between the Problem attributes and various LLM Services. We use two prompts for our experiments, one baseline, and one expanded to further prompt the LLM Service to conform to a consistent symbolic representation.

**Algorithm 1** Problem Initialization

**Require:** Sampling configuration (distributions for variables, constraints, objectives), desired semantic problem type, solver parameters, and optionally resource functions.
**Ensure:** Symbolic problem, natural language description, solver-compatible code, and solver output.
 1: Sample number of variables using multi-branch distribution
 2: Sample number of resources using hypergeometric distribution
 3: Generate resource matrix and budget caps and store.
 4: Determine problem type by maximum Degree, maximum height, Variable type (continuous, mixed, or integer), and Goal (maximize or minimize)
 5: Generate term space for objective function
 6: Construct symbolic objective from randomly sampled monomials and coefficients
 7: Generate constraint set by : 1) Non-negativity, integrality, upper and lower bounds including tradeoff and budget constraints
 8: Assign semantic problem type based on variable type.
 9: Sample variable names and resource descriptions from semantic type
10: Generate natural language mappings for Variables, Resources, Constraints, and Objective function, and store.
11: Compile symbolic and natural language problem descriptions
12: Translate to solver-executable code and store (e.g., Gurobi)
13: Run solver-executable code and store solution
14: Return Problem object

We use the following as to generate our two prompts, with the condition `include_sym` to trigger the expanded prompt:

**Prompt Template**

```
"PURPOSE: I need you to solve an optimization problem,
outputting Gurobi code that captures the problem description
and provides a solution, or otherwise indicates the problem
is infeasible.
CONTEXT: I have the following variables to consider:
{problem.problem_variables} which have the following
resources/attributes that I need to deal with:
{problem.nl_resources_dict}
ROLE: You are a consulting team of business analysts,
operations researchers, and programmers who will convert my
natural language description of an optimization problem into
functional Gurobi code that answers my problem.
INPUT: I need to {problem.goal} the following objective
function : {problem.objective_function_statement} subject
to the following constraints:"
for constraint in problem.problem_constraints:  "*
{constraint}"
"OUTPUT:"
if include_sym:  "In order to convince me that the code you
are producing is correct, I also need to have a symbolic
representation of the problem showing me that you have
converted the description above into an appropriate symbolic
representation of the optimization problem.  This consists of
a pairs of variables in symbolic notation for the first item
in the pair of the form 'x1', 'x2', and so on, and the second
item of the pair being the natural language object appearing
in the problem description; the objective function rendered
as an algebraic term where all natural language objects are
substituted for the corresponding symbolic variable; and
the list of semi-algebraic constraints where the natural
language object is substituted with its symbolic variable
counterpart.  Return this solution in a code bloc encased
as ```json {dict({"sym_variables":  [("x#i","object#i")],
"objective_function":"objective function description with sym
variables", "constraints":["constraint",]})} ``` " fi
```

**Prompt Template (Cont'D)**

```
" Finally, please output Gurobi code enclosed as ```python

```.
* Do not have anything else AFTER this final block.
* If you provide any reasoning for your final answer, you
MUST put it before the final ```python  ``` bloc "
```

## C.3 SYMINTERCHANGE

The **SymInterchange** class was designed to produce 'inverse' methods for the **Problem** class, and produce an IR, intended for ingestion by an Agent or teams of Agents. The class instantiates an intermediate data structure for representing optimization problems in both natural language and symbolic mathematical form, with methods starting from the natural language representation.

The class is initialized with a dictionary of key attributes such as the problem statement, variables, resources, constraints, and objective function, along with metadata like the problem type and goal. It provides methods to extract symbolic representations from natural language descriptions, convert

objectives and constraints into symbolic form, and optionally generate executable code (e.g., for Gurobi) to solve the problem.

The class also includes functionality for loading, saving, updating, and verifying the consistency of these symbolic-natural language mappings, enabling seamless transformation between interpretable and computable representations of structured optimization tasks. We have included it in the Supplementary Files for researchers.

# D    DATA

### D.0.1    NLMOPTIMIZER DATA

We have included 1000 problems generated by the Problem class. We have included both the Problem class as well as the problems in our Supplementary files. Beyond simple slot-filling, our templates incorporate anaphora, ellipsis, proportional statements, and domain jargon, in line with other efforts such as Xiao et al. (2024) each pattern family is listed with multiple surface variants and examples to support inspection and reuse.

Each generated problem is paired with a natural language representation. This is achieved by inserting the parameters of the formal optimization model into pre-built natural language templates. These templates are structured to mimic human-authored prompts, and include redundant sentence options to introduce variation to natural language representations. Natural language prompts are assembled from templates using a combination of string formatting and regular expression substitution, ensuring that the resulting language is coherent and lexically diverse, while remaining traceable to the underlying mathematical representation of an optimization problem.

The generation of natural language representations begins with the random selection of a problem context, `semantic_type`. The value of `semantic_type` determines the possible values for resources, values, and some template text, ensuring that generated text is context-consistent and grammatically correct. An initial batch of available contexts covers: Office supply budgeting, Household budgeting, Horticulture, Cybersecurity, staffing and management, Military force structure, Personnel management, Personal diet, and Macronutrient intake.

A natural language statement is selected from a set of alternative wordings to describe the problem context, which places the agent in the role of the solver. Variables and resources are assigned labels selected at random from sets of available labels for the selected context. Statements describing resource costs for each variable are generated after randomly selecting from a set of context-relevant sentence templates, before substitution with the corresponding variable names, resource names, and resource costs. For any non-negativity constraints, statements are generated by randomly selecting from multiple sets of sentence fragments and joining them to produce a single grammatically-correct sentence asserting non-negativity. Upper- and lower-bound constraint statements are then generated by random selection from a context-relevant set of available sentence templates, followed by term substitution with the relevant resource and variable names. We consider proportion constraints to be part of the lower-bound constraints.

We denote by `int_type` whether the optimization problem is integer, mixed-integer, or linear. `int_type` is a randomly determined by two initializing parameters that partition the unit interval. This was done so that users may opt to force a particular problem type if they so desired. The `int_type` determines the statements that can be generated where the permissible solution value types are described. As with other methods, these statements are formed by combining selecting randomly from multiple sets of sentence fragments and joining them to produce grammatically-correct sentences conditioned on the `int_type`. Similarly, the natural language objective function statement is generated by first iterating through the symbolic objective function, and substituting variables, values, and operators with their previously selected semantic equivalents in natural language, before randomly selecting from a set of sentence fragment templates that are used to construct a full natural language statement. Finally, we collect all of the statements generated by the above methods are collected into a single problem statement.

We illustrate this process with the following example. Consider the case of a 2-variable 1-resource integer optimization problem with a goal of maximization, and objective function: $2 \cdot x_0 + 3 \cdot x_1 + x_0 \cdot x_1$ and three constraints:

1. $x_0 \geq 0$
2. $x_1 \geq 0$
3. $x_0 + x_1 \leq 10$

Once the symbolic representation of the problem has been generated, the context `diet0` (personal diet) is selected at random.

```python
def _select_semantic_problem(self):
    """

    Selects a semantic problem type for determining.
    Supports up to 12 vars.
    """

    if self.problem_type["is_integer"] == 1:  # Integer
        semantic_problem_options = [
            "office supplies",
            "family budget",
            "gardening",
            "network defense",
            "force structure",
            "diet0",
            "personnel",
        ]

    elif self.problem_type["is_integer"] == 0.5:  # Mixed integer
        semantic_problem_options = [
            "diet0",
            "diet1",
            "personnel",
        ]

    elif self.problem_type["is_integer"] == 0:  # continuous/linear
        semantic_problem_options = [
            "diet0",
            "diet1",
            "personnel",
        ]

    return random.choice(semantic_problem_options)
```

Figure 12: Method to select `semantic_type`.

A dictionary mapping variables to names, `sym_vars`, is constructed by selecting a random sample of size `num_vars` from the diet0 context-relevant list of possible variable names.

```
elif self.semantic_problem_type == "diet0":
    possible_semantic_vars = [
        "eggs",
        "apple pies",
        "cherry pies",
        "blueberry pies",
        "chicken breasts",
        "chicken thighs",
        "chicken drumsticks",
        "pickles",
        "kale salads",
        "fruit salads",
        "apples",
        "lemons",
        "hamburgers",
        "cheeseburgers",
        "rotisserie chickens",
        "steaks",
        "ravioli",
        "milkshakes",
```

Figure 13: A portion of the `possible_semantic_vars` list.

```
# Build sym vars
sym_vars = dict()
for i in range(self.num_vars):
    var = f"x{i}"
    varname = selected_vars[i]
    vartokens = tokenized_vars[varname]
    common_objects = [
        token for token in vartokens if token_counts[token] > 1
    ]  # all repeated tokens
    differentiators = [
        token for token in vartokens if token not in common_objects
    ]  # all non-repeated tokens
    # Only keep differentiators if there's at least one common object
    if not common_objects:
        differentiators = []

    item_dict = {
        "full_phrase": varname,
        "differentiators": differentiators,
        "common_objects": common_objects,
        "target": None,
    }
    sym_vars[var] = item_dict

return sym_vars
```

Figure 14: Method to produce the `sym_vars` dict.

For our example, suppose that "apples" and "bananas" have been randomly selected. Likewise, a dictionary mapping resources to names, nl_resources_dict, is constructed by selecting a random sample of size num_resources from the diet0 context-relevant list of possible resource names.

```python
elif self.semantic_problem_type == "diet0":
    possible_resources = [
        "dollar cost",
        "grams of protein",
        "grams of carbohydrates",
        "grams of fat",
        "grams of fiber",
        "milligrams of calcium",
        "milligrams of iron",
        "tastiness rating",
        "healthiness rating",
        "sourness index",
        "umami index",
    ]
```

Figure 15: A portion of the possible_resources list.

```python
selected_resources = random.sample(possible_resources, self.num_resources)
nl_resources_dict = dict()
for i in range(self.num_resources):
    r = f"r{i}"
    item_dict = {
        "description": selected_resources[i],
        "upper_bound": self.resources_dict["budget_caps"][i],
    }
    for j in range(self.num_vars):
        var = f"x{j}"
        weight = self.resources_dict["weights"][j, i]
        item_dict[var] = weight
    nl_resources_dict[r] = item_dict

return nl_resources_dict
```

Figure 16: Method to produce nl_resources_dict.

For our example, say we select "milligrams of calcium". Next we want to generate value statements for each variable and resource. We randomly select a template from available options for each resource and input names and values to produce the full statements.

```python
elif resource_name in [
    "grams of protein",
    "grams of carbohydrates",
    "grams of fat",
    "grams of fiber",
    "milligrams of calcium",
    "milligrams of iron",
]:
    weight_statements = [
        f"{varname} each contain {weight} {resource_name}. ",
        f"{varname} contain {weight} {resource_name}. ",
        f"There are {weight} {resource_name} in {varname}. ",
    ]
```

Figure 17: Weight statement options for milligrams of calcium.

One possible output would be: "apples each contain 2 milligrams of calcium." Non-negativity constraints are then generated by selecting sentence fragments, joining them, and inputting variable names.

```
for constraint_name, constraint in self.constraints["non_negativity"].items():
    for var in self.sym_vars.keys():
        varname = self.sym_vars[var]["full_phrase"]
        if var + " >= 0" in constraint:
            if random.random() > 0.5:  # Positive statement
                imperatives = [
                    "You must use ",
                    "You must have ",
                    "You have to use ",
                    "You have to have ",
                    "There must be ",
                    "There has to be ",
                ]
                operator_statements = [
                    "no less than zero ",
                    "no less than 0 ",
                    "zero or more ",
                    "0 or more ",
                    "zero or greater ",
                    "0 or greater ",
                    "greater than or equal to zero ",
                    "greater than or equal to 0 ",
                    "a non-negative number of ",
                    "no less than zero ",
                    "no less than 0 ",
                ]
                sym_constraint = (
                    random.choice(imperatives)
                    + random.choice(operator_statements)
                    + varname
                    + ". "
                )
```

Figure 18: Non-negativity statement segment construction.

For instance, this could produce the statement: "You must use no less than zero apples." Upper- and lower-bound statements are assembled by inputting the appropriate values into templates.

```python
elif resource in [
    "grams of carbohydrates",
    "grams of fat",
    "grams of fiber",
    "milligrams of calcium",
    "milligrams of iron",
]:
    ub_statement = random.choice(
        [
            f"You must get at most {bound} {resource} of from {joined_vars}. ",
            f"You must get no more than {bound} {resource} of from {joined_vars}. ",
            f"You need to get no more than {bound} {resource} of from {joined_vars}. ",
            f"At most {bound} {resource} can come from {joined_vars}. ",
            f"You cannot get more than {bound} {resource} from {joined_vars}. ",
            f"You can get up to {bound} {resource} from {joined_vars}. ",
        ]
    )
```

Figure 19: Upper-bound statement segment construction.

In the instance of our example upper bound constraint, this may generate the statement "You must get at most 10 milligrams of calcium from apples and bananas." Integer constraints will then be generated for each variable by joining sentence fragment templates with resource names, similar to non-negativity constraints.

```python
if constraint["bound"] == "int":
    if random.random() > 0.33:
        imperatives = [
            "You must use ",
            "You have to use ",
            "You must have ",
            "You have to have ",
            "There must be ",
            "There has to be ",
            "You are restricted to ",
            "You're restricted to ",
            "You are limited to ",
            "You're limited to ",
        ]
        int_statements = [
            "a whole number of ",
            "a whole number amount of ",
            "an integer number of ",
            "an integer amount of ",
            "a non-fractional amount of ",
            "a nonfractional number of ",
        ]
        sym_constraints[f"{constraint_name}"] = (
            random.choice(imperatives)
            + random.choice(int_statements)
            + varname
            + ". "
        )
```

Figure 20: Integer type statement segment construction.

This may produce the statement: "You must use an integer amount of apples." For the objective function, we again select randomly from multiple sentence fragments to start the statement, then loop over the terms in the objective function to produce a natural language representation of the entire function and join that to our initial sentence fragment to produce the full statement.

```python
def _gen_objective_function_statement(self):

    objective_function_statement = random.choice(
        [
            f"Your overall goal is to {self.goal} ",
            f"Your goal is to {self.goal} ",
            f"You need to {self.goal} ",
            f"You need to {self.goal} the value of ",
            f"We need to {self.goal} ",
            f"We need to {self.goal} the value of ",
            f"Altogether, we want to {self.goal} ",
        ]
    )

    for vartuple in self.objective_function_data["term_list"]:
        if not isinstance(vartuple[0], str):
            coef = round(vartuple[0], 2)
            coef = str(coef)
        operator = random.choice(
            [
                " times the number of ",
                " multiplied by the number of ",
                " times the amount of ",
                " multiplied by the amount of ",
                " times the quantity of ",
                " multiplied by the quantity of ",
                " times the total number of ",
                " times the quantity of ",
            ]
        )
        varname = self.get_varname(vartuple[1])
        trailing_operator = random.choice([" plus ", " added to "])
        objective_function_statement += (
            coef + operator + varname + trailing_operator
        )
    objective_function_statement = (
        objective_function_statement[: -len(trailing_operator)] + ". "
    )

    return objective_function_statement
```

Figure 21: Method to produce natural language objective function statements.

For instance: "Your overall goal is to maximize 2 times the number of apples plus three times the number of bananas plus apples times the number of bananas." Lastly, all of the above will be joined into a singular statement. This begins by selecting a context-relevant initial problem statement establishing context.

```
elif self.semantic_problem_type == "diet0":
    problem_statement = random.choice(["I need you to help me develop a diet plan that provides me the correct balance of nutrients and other metrics. ",
                                       "Help me refine my diet plan to provide the right balance of nutrients and other metrics. ",
                                       "I need help determining a diet plan that balances nutrients and other metrics correctly. ",
                                       "My diet is unbalanced, and I need to find a way to adjust my intake of several foods to get the right mix of nutr",
                                       "I need to optimize my nutrient intake. "])
```

Figure 22: Problem statement initial sentence list.

Variable and resource statements are then appended iteratively.

```
# Add sym vars
problem_statement += random.choice(["These are the foods that I need to adjust my intake of: ",
                                    "These are all of the foods that need to be balanced in my diet: ",
                                    "These are the foods from which I need to get all of my nutrients: ",
                                    "You need to find the right intake of ",
                                    "I eat ",
                                    "The only things I eat are ",
                                    "All I eat are ",
                                    "I only eat "
                                    "My diet must consist entirely of "])
i = 0
for var in self.problem_variables:
    if i < self.num_vars-2:
        problem_statement += var + ", "
    elif i == self.num_vars-1:
        problem_statement += var + random.choice([" and ", ", and "])
    elif i == self.num_vars:
        problem_statement += var + ", "
    i+=1

# Add sym resources
problem_statement += random.choice(["I need to get the right balance of ",
                                    "I need to optimize my intake of ",
                                    "I need to balance my ",
                                    "I care about optimizing "])
i = 1
for res in self.resources:
    if i < self.num_resources-2:
        problem_statement += res + ", "
    elif i == self.num_resources-1:
        problem_statement += res + random.choice([" and ", ", and "])
    elif i == self.num_resources:
        problem_statement += res + ". "
    i+=1
```

Figure 23: Problem statement initial sentence list.

And finally, each constraint statement and the objective statement are simply appended to the problem_statement string.

```
# Add sym vars
problem_statement += random.choice(["These are the foods that I need to adjust my intake of: ",
                                    "These are all of the foods that need to be balanced in my diet: ",
                                    "These are the foods from which I need to get all of my nutrients: ",
                                    "You need to find the right intake of ",
                                    "I eat ",
                                    "The only things I eat are ",
                                    "All I eat are ",
                                    "I only eat "
                                    "My diet must consist entirely of "])
i = 0
for var in self.problem_variables:
    if i < self.num_vars-2:
        problem_statement += var + ", "
    elif i == self.num_vars-1:
        problem_statement += var + random.choice([" and ", ", and "])
    elif i == self.num_vars:
        problem_statement += var + ". "
    i+=1

# Add sym resources
problem_statement += random.choice(["I need to get the right balance of ",
                                    "I need to optimize my intake of ",
                                    "I need to balance my ",
                                    "I care about optimizing "])
i = 1
for res in self.resources:
    if i < self.num_resources-2:
        problem_statement += res + ", "
    elif i == self.num_resources-1:
        problem_statement += res + random.choice([" and ", ", and "])
    elif i == self.num_resources:
        problem_statement += res + ". "
    i+=1
```

Figure 24: Problem statement initial sentence list.

The fully constructed problem statement is then saved to the instantiated `Problem` object. In our example case, this would evaluate to: "I need to optimize my nutrient intake. My diet must consist entirely of apples and bananas. I need to optimize my intake of milligrams of calcium. Apples contain 2 milligrams of calcium. Bananas contain 3 milligrams of calcium. You must use no less than zero apples. You must use a positive number of bananas. You must get at most 10 milligrams of calcium from apples and bananas. You must use an integer amount of apples. You must use a whole number of bananas. Your overall goal is to maximize 2 times the number of apples plus three times the number of bananas plus apples times the number of bananas."

## E    ADDITIONAL EXPERIMENTAL STATISTICS

Commenting first on Table 1, we display a model and prompt breakdown across five categories: end-to-end Accuracy, Adjusted end-to-end Accuracy, Compilation Error, Runtime Error, and Adjusted Runtime Error. The adjusted end-to-end Accuracy and run time scores apply to allow model outputs after performing a regex search to identify if there was a failure to import Gurobi properly, which required `import gurobipy`. Failures frequently resulted from attempts to run the incorrect expression: `import gurobi`. This happened overwhelmingly with LLaMa-4, and presents

Table 3: Summary of problems generated by integer, mixed integer, and continuous linear & quadratic programs

|          | ILP  | IQP  | MILP | MIQP | LP   | QP   |
|----------|------|------|------|------|------|------|
| med Vars | 4    | 3    | 4    | 4    | 4    | 4    |
| Max Vars | 8    | 7    | 9    | 7    | 9    | 8    |
| med Res  | 2    | 2    | 2    | 2    | 2    | 2    |
| Max Res  | 5    | 5    | 5    | 5    | 5    | 5    |
| med Cons | 33   | 27   | 32   | 32   | 32   | 36   |
| Max Cons | 1448 | 1021 | 1994 | 635  | 2009 | 1977 |
| Feas     | 79   | 12   | 81   | 16   | 164  | 22   |
| Total    | 184  | 47   | 198  | 113  | 372  | 86   |

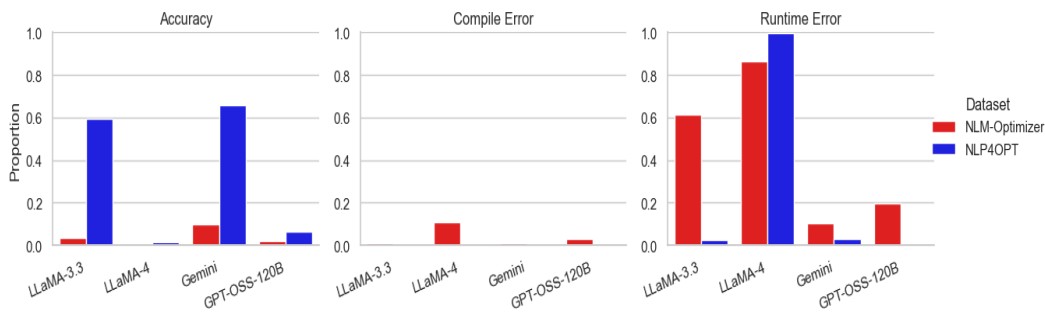

Figure 25: Summary results for each model on NL4OPT dataset vs NLMOptimizer dataset.

a curiosity more than anything, as even after this adjustment, LLaMa-4 overwhelmingly suffered from runtime errors. LLaMa-4 also overwhelmingly produced compilation errors when compared with the other models. Successful solutions include both cases where the model produced correct executable Gurobi code and cases where the model correctly identified the problem as infeasible.

In all Figures where models are end-labeled by '_op_result', we mean to indicate model results for the original problems produced for these experiments, with the additional end-label of '_wsym' indicating the expanded prompting with additional prompting for symbolic representation.

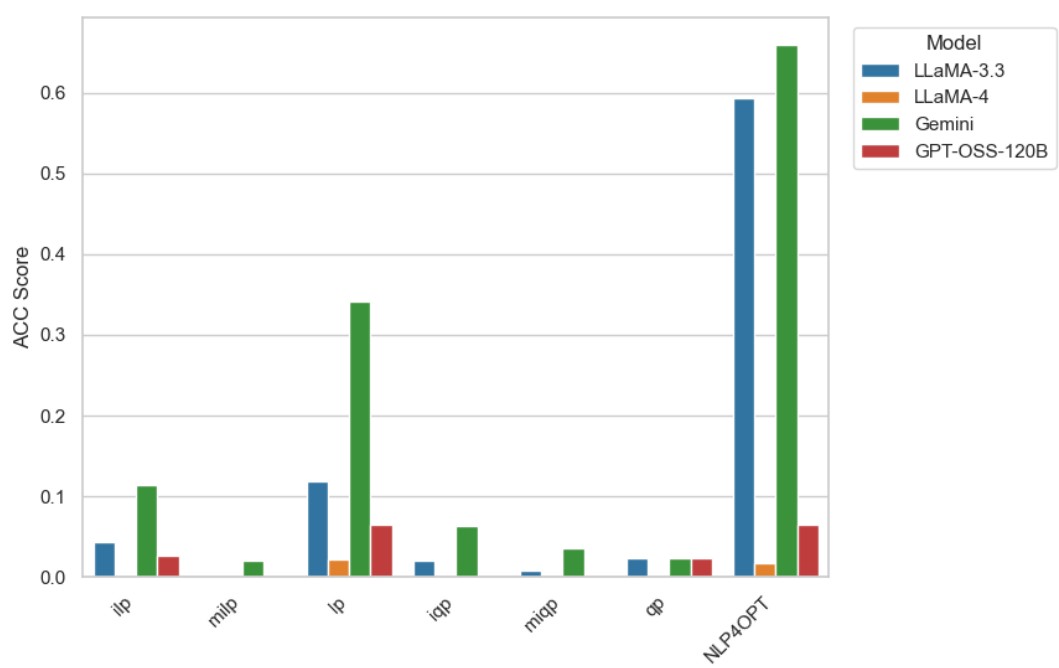

Figure 26: End-to-End accuracy scores by model and problem type.

Overall, the poor performance of LLaMa-4 applied to both datasets and both prompts. More interesting for our purposes is the collapse in performance for the other models when tasked with answering the NLMOptimizer problems. In Tables 4 to 10, we display the end-to-end Accuracy, Compilation Error, and Runtime Error by each model, prompt, and problem type. We collectively visualize the end-to-end Accuracy, Compilation Error, and Runtime Errors across model, prompt, and model type in Figures 26 – 31.

We consistently observe that the model performance collapses for any problem that is not a standard linear program. Most especially, we observe that even Gemini-1.5 suffers on the linear programs

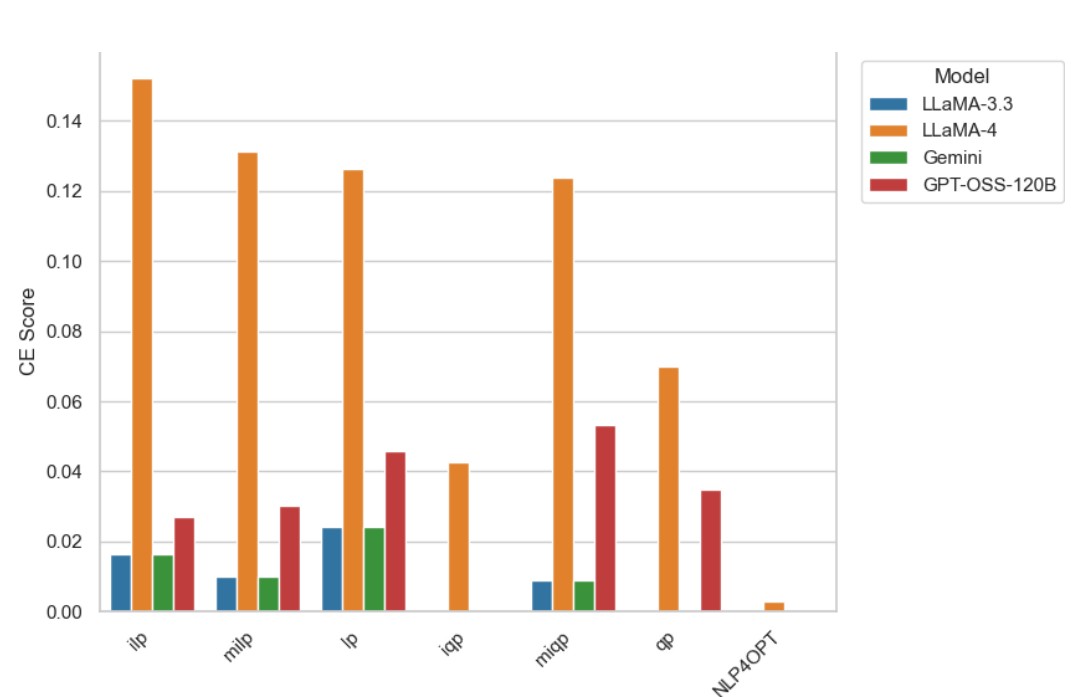

Figure 27: Compilation error scores by model and problem type.

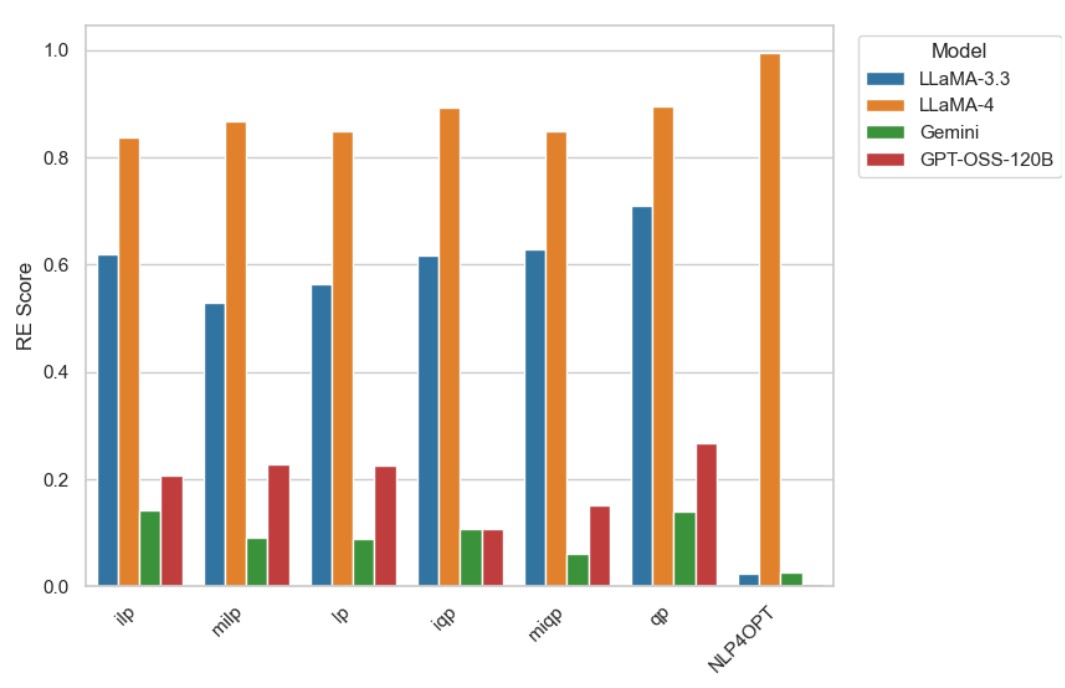

Figure 28: Runtime error scores by model and problem type.

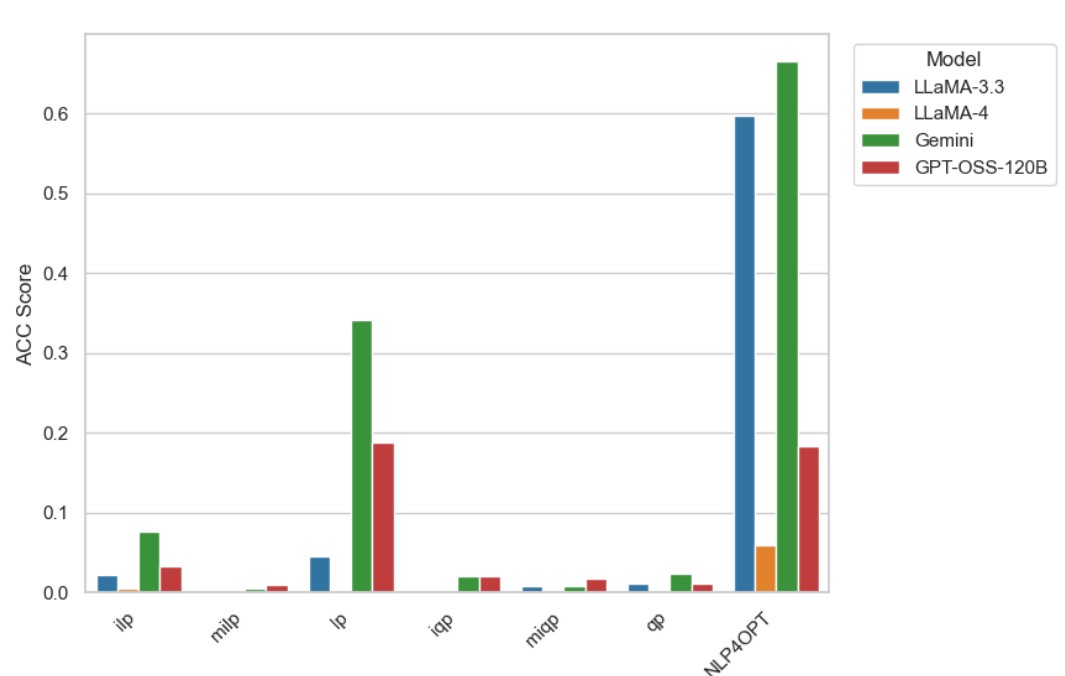

Figure 29: End-to-end accuracy scores by model with additional prompt and problem type.

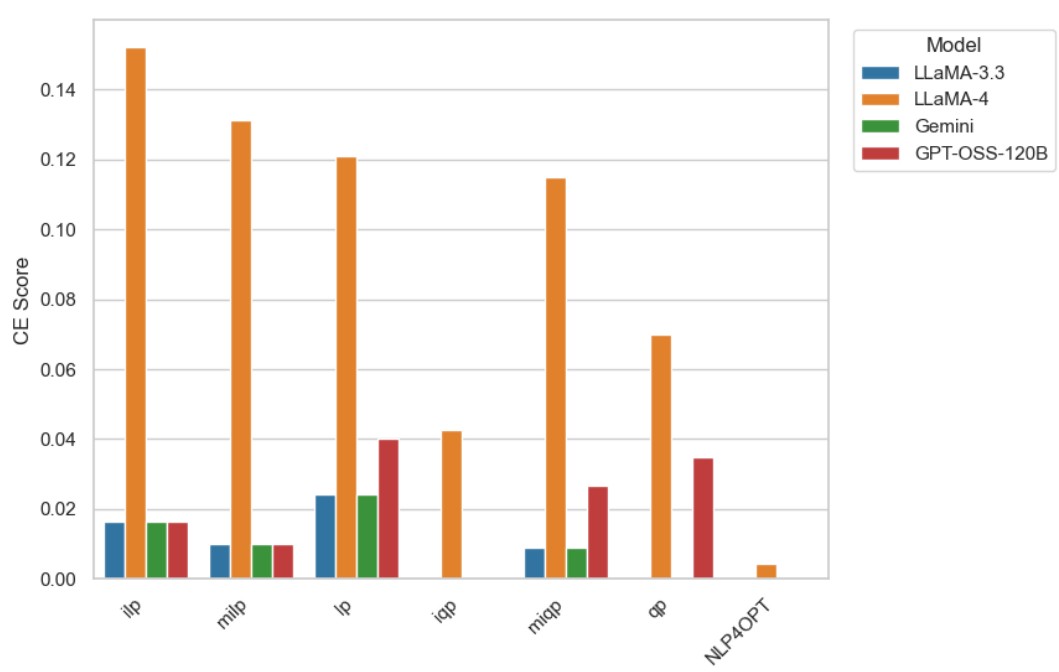

Figure 30: Compilation error scores by model with additional prompt and problem type.

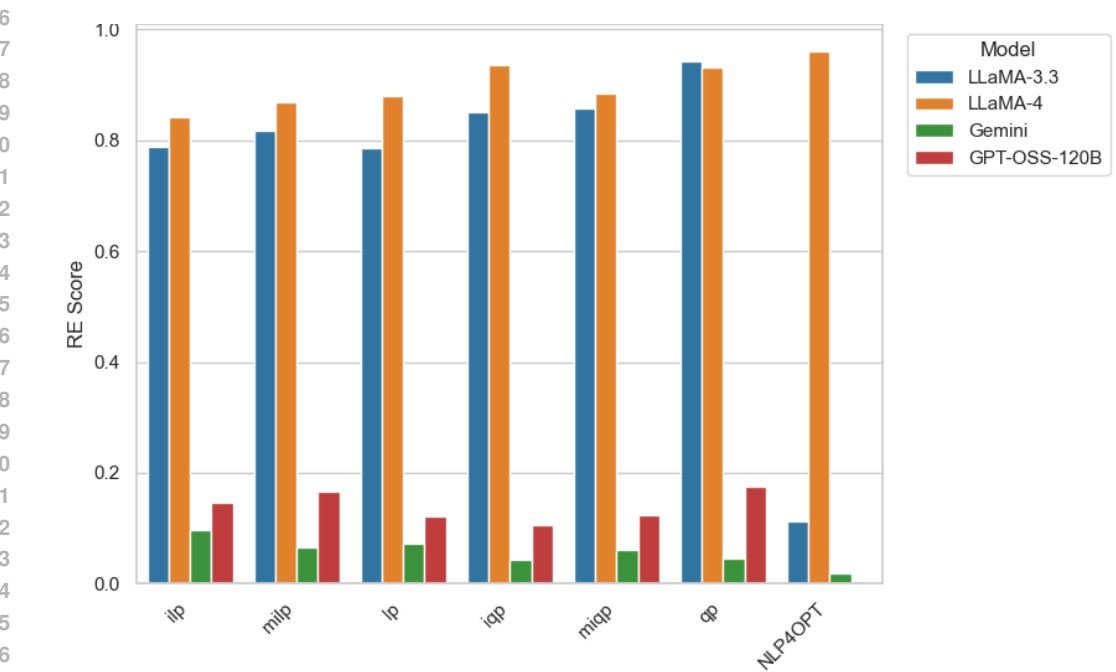

Figure 31: Runtime error scores by model with additional prompt and problem type.

Table 4: Problem type statistics for LLaMa-3.3 and NLMOptimizer problems

|      | ACC   | CE    | RE    |
|------|-------|-------|-------|
| ilp  | 0.043 | 0.016 | 0.620 |
| milp | 0.000 | 0.010 | 0.530 |
| lp   | 0.118 | 0.024 | 0.565 |
| iqp  | 0.021 | 0.000 | 0.617 |
| miqp | 0.009 | 0.009 | 0.628 |
| qp   | 0.023 | 0.000 | 0.709 |

Table 5: Problem type statistics for LLaMa-4 and NLMOptimizer problems

|      | ACC   | CE    | RE    |
|------|-------|-------|-------|
| ilp  | 0.000 | 0.152 | 0.837 |
| milp | 0.000 | 0.131 | 0.869 |
| lp   | 0.022 | 0.126 | 0.849 |
| iqp  | 0.000 | 0.043 | 0.894 |
| miqp | 0.000 | 0.124 | 0.850 |
| qp   | 0.000 | 0.070 | 0.895 |

Table 6: Problem type statistics for Gemini and NLMOptimizer problems

|      | ACC   | CE    | RE    |
|------|-------|-------|-------|
| ilp  | 0.114 | 0.016 | 0.141 |
| milp | 0.020 | 0.010 | 0.091 |
| lp   | 0.341 | 0.024 | 0.089 |
| iqp  | 0.064 | 0.000 | 0.106 |
| miqp | 0.035 | 0.009 | 0.062 |
| qp   | 0.023 | 0.000 | 0.140 |

Table 7: Problem type statistics for GPT-OSS and NLMOptimizer problems

|      | ACC   | CE    | RE    |
|------|-------|-------|-------|
| ilp  | 0.027 | 0.027 | 0.207 |
| milp | 0.000 | 0.030 | 0.227 |
| lp   | 0.065 | 0.046 | 0.226 |
| iqp  | 0.000 | 0.000 | 0.106 |
| miqp | 0.000 | 0.053 | 0.150 |
| qp   | 0.023 | 0.035 | 0.267 |

Table 8: Problem type statistics for LLaMa-3.3 and NLMOptimizer problems and expanded prompt

|      | ACC   | CE    | RE    |
|------|-------|-------|-------|
| ilp  | 0.022 | 0.016 | 0.788 |
| milp | 0.000 | 0.010 | 0.818 |
| lp   | 0.046 | 0.024 | 0.785 |
| iqp  | 0.000 | 0.000 | 0.851 |
| miqp | 0.009 | 0.009 | 0.858 |
| qp   | 0.012 | 0.000 | 0.942 |

Table 9: Problem type statistics for LLaMa-4 and NLMOptimizer problems and expanded prompt

|      | ACC   | CE    | RE    |
|------|-------|-------|-------|
| ilp  | 0.005 | 0.152 | 0.842 |
| milp | 0.000 | 0.131 | 0.869 |
| lp   | 0.000 | 0.121 | 0.879 |
| iqp  | 0.000 | 0.043 | 0.936 |
| miqp | 0.000 | 0.115 | 0.885 |
| qp   | 0.000 | 0.070 | 0.930 |

Table 10: Problem type statistics for Gemini and NLMOptimizer problems and expanded prompt

|      | ACC   | CE    | RE    |
|------|-------|-------|-------|
| ilp  | 0.076 | 0.016 | 0.098 |
| milp | 0.005 | 0.010 | 0.066 |
| lp   | 0.341 | 0.024 | 0.073 |
| iqp  | 0.021 | 0.000 | 0.043 |
| miqp | 0.009 | 0.009 | 0.062 |
| qp   | 0.023 | 0.000 | 0.047 |

Table 11: Problem type statistics for GPT-OSS and NLMOptimizer problems and expanded prompt

|      | ACC   | CE    | RE    |
|------|-------|-------|-------|
| ilp  | 0.033 | 0.016 | 0.147 |
| milp | 0.010 | 0.010 | 0.167 |
| lp   | 0.188 | 0.040 | 0.121 |
| iqp  | 0.021 | 0.000 | 0.106 |
| miqp | 0.018 | 0.027 | 0.124 |
| qp   | 0.012 | 0.035 | 0.174 |

generated under the NLMOptimizer framework, with end-to-end Accuracy below .35. Gemini-1.5 also managed to be inaccurate, but not due to code failure, whereas the LLaMa models both had high incidence rates of run time errors, with LLaMa-4 also suffering from high compilation errors above 10% for every problem type but quadratic programs and integer quadratic programs. Conversely, the LLaMa models consistently had trouble with all forms of quadratic programs, with Runtime Errors above 80% as seen in Figure 28.

We then consider a granular breakdown across our three categories by the number of problem variables and problem types, with each model and prompt pairing being visualized in its own respective figure (across Figures 4-11). Specifically, in Figure 4, we observe that LLaMa-3.3 performs poorly for all problem types, with scores generally worsening as the number of variables increases across problem types, although curiously end-to-end Accuracy scores are parabolic for classical linear programs, and bottom out at 5 variables. In contrast, Figure 8 shows that the end-to-end Accuracy does not exhibit this parabola when expanding the prompt to include symbolic reasoning. The general trend for both prompts is that compile and runtime errors increase as the number of variables increases across all problem types. Figures 5 and 9 show that LLaMa-4 remained a poor performer across all categories and all problem types, under-performing even in comparison to its own predecessor model.

Token limitation for LLaMa-3.3 ruled out all but the simplest linear and quadratic programs whenever the full conversation history was provided as context. We also found that the GPT-OSS-120B model performed poorly on both datasets, but demonstrated significant improvements when given additional prompting to follow the symbolic interchange format. Removing either GPT-OSS-120B or the Gemini model eliminated the significance of the E2a results, which were otherwise insignificant with respect to the two LLaMa models.

We breakdown Gemini-1.5's performance with respect to the number of variables in Figures 6 and 11. Again, general trends indicate that performance declines as problem complexity increases across all problem types. Linear programs remain the best performers. Curiously, Gemini-1.5's performance on integer linear programs shot up from 0% to almost 50% when the number of variables increased from 7 to 8, despite having bottomed out at 6 variables. Crucially, we witnessed that runtime errors increased in proportion to the number of variables across all models.

Finally, we also broke down the performance of each model and prompt pair according to whether the problem was feasible by problem type in Figures 32 - 38. We generally observed that models performed more 'accurately' when the problem was unsolvable than when it was solvable across all problem types. We suspect this can be attributed to our measurement method, which simply examined that the output of the problem matched the corresponding Problem output; it is infinitely easier to make a problem infeasible. Consider an LLM Service incorrectly recording one constraint as another- both the correct constraint and the incorrect constraint happen to render a problem description infeasible. In such a case, the LLM Service might get the correct answer unintentionally, despite incorrectly formalizing the problem.

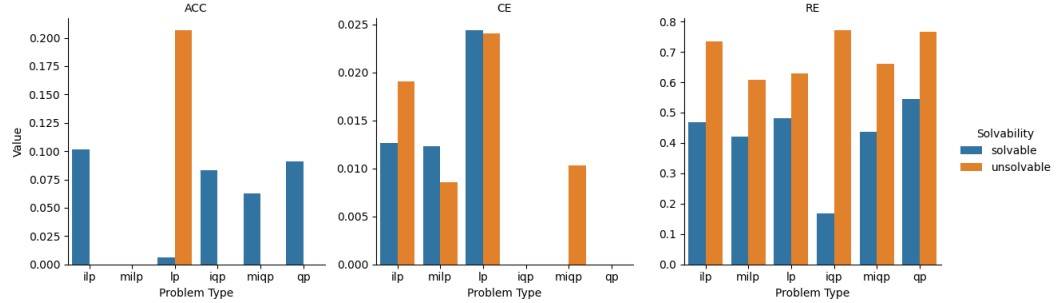

Figure 32: Distribution of scores for LLaMa-3.3 according to problem feasibility.

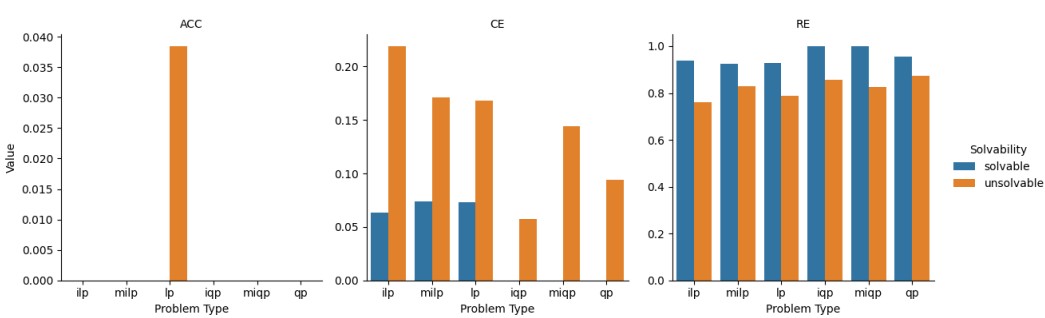

Figure 33: Distribution of scores for LLaMa-4 according to problem feasibility.

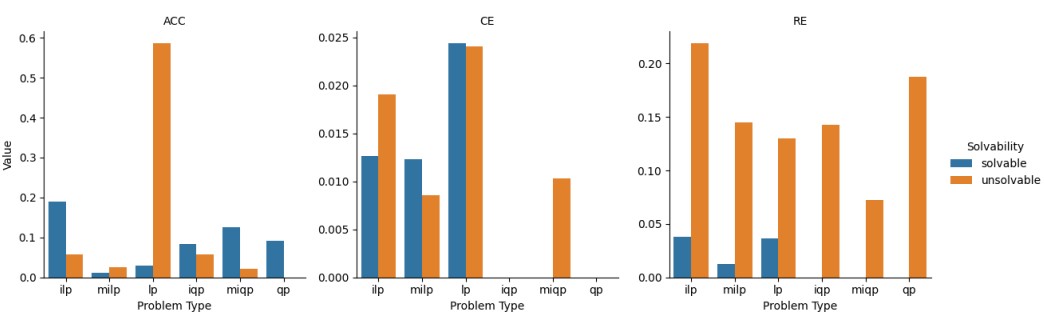

Figure 34: Distribution of scores for Gemini according to problem feasibility.

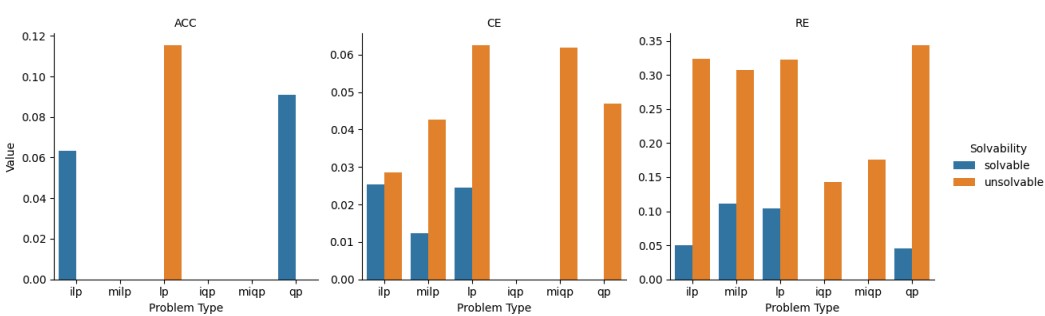

Figure 35: Distribution of scores for GPT-OSS according to problem feasibility.

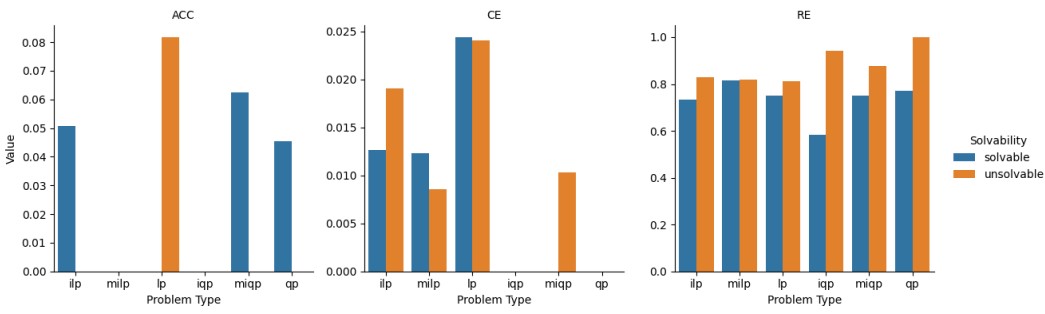

Figure 36: Distribution of scores for LLaMa-3.3 with additional prompting according to problem feasibility.

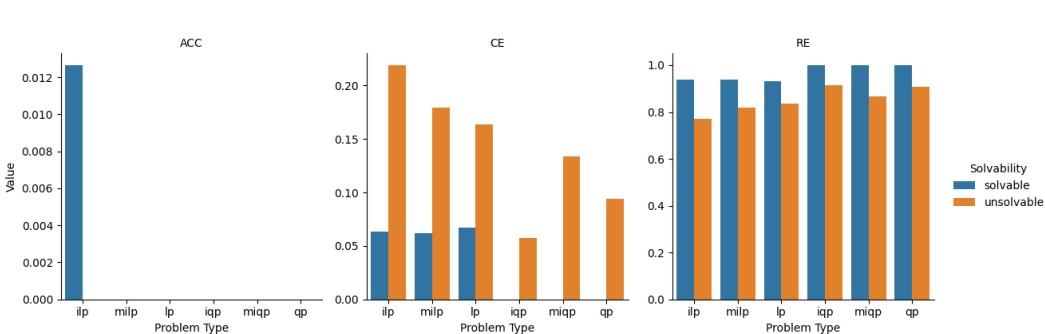

Figure 37: Distribution of scores for LLaMa-4 according to problem feasibility.

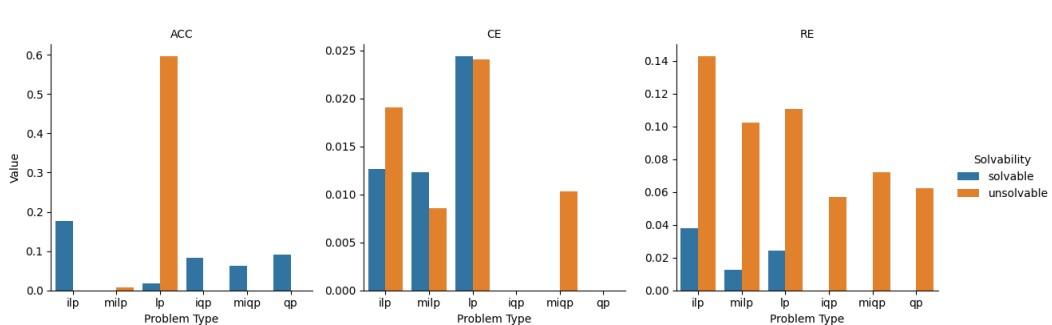

Figure 38: Distribution of scores for Gemini according to problem feasibility.

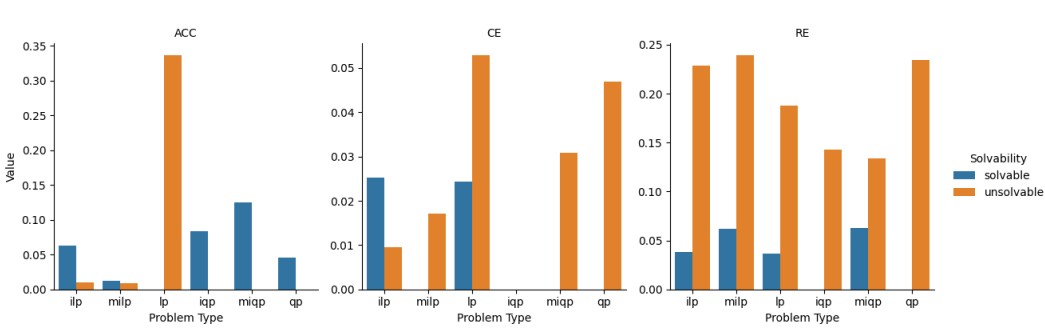

Figure 39: Distribution of scores for GPT-OSS according to problem feasibility.

