# OpenReview forum: "NLMOptimizer: A neurosymbolic framework and benchmark for operations research optimization problems from natural language"
_ICLR.cc/2026/Conference — ICLR 2026 Conference Withdrawn Submission_

### Official Review · Reviewer_6CFM · 2025-10-30

**Soundness:** 3
**Presentation:** 2
**Contribution:** 2
**Rating:** 4
**Confidence:** 4

**Summary:**

This paper proposes a neurosymbolic framework, NLMOptimizer, and compares the performance of four instruction-tuned LLMs.

**Strengths:**

- SymInterchange is a novel representation method that maps problems described in natural language to the form of solvers.
- The authors considered a variety of optimization problems, rather than being limited to MILP problems, although I only found the NL4Opt dataset in the code.

**Weaknesses:**

- As shown in Figure 3, the authors use the number of variables to represent the difficulty of the problem. Is it possible that with the same 10 variables, a simple MILP problem might not be very difficult to solve, but if the problem becomes a combination optimization problem with complex constraints and objectives, the difficulty increases dramatically?
- Is 10 a small number of variables, insufficient to represent a complex problem? For example, a MILP problem with more than 100 variables.
- For optimization problems with multiple unique modeling approaches, how can SymInterchange represent different modeling methods? And how would this affect the effectiveness of the framework?
- What is the potential of NLMOptimizer in solving combinatorial optimization problems? For example, how many nodes can it solve in a Traveling Salesman Problem?
- The field of LLM for solving optimization problems has developed rapidly. Benchmark work should include a comprehensive comparison of related methods. For example, LLMOPT, OptMATH, SIRL, and ORLM, as excellent post-training methods in this field, should all be evaluated.
- The details of the paper should be carefully checked, such as the fact that `LLM` is written as lowercase `llm` in many places in the references.

**Questions:**

As described in weaknesses.

---

### Official Review · Reviewer_jRxT · 2025-10-31

**Soundness:** 2
**Presentation:** 2
**Contribution:** 2
**Rating:** 2
**Confidence:** 3

**Summary:**

This paper introduces NLMOptimizer, a neurosymbolic framework and benchmark for translating natural language descriptions of operations research (OR) optimization problems into formal, solver-executable mathematical programs. The work addresses a critical gap in existing benchmarks, which are often too simplistic to reflect real-world complexity, and highlights the limitations of current large language models (LLMs) in accurately representing and solving such problems.

**Strengths:**

* **Better Benchmark**: Creates a more realistic and challenging test (NLMOptimizer) that exposes the true difficulty of the task, where even top LLMs fail.
* **Clear Insight**: Correctly identifies the core problem as a failure in representation, not just code generation, shifting the focus for future research.

**Weaknesses:**

* **Insufficient Benchmark Comparison**: The paper positions its NLMOptimizer dataset as a core contribution, yet provides a limited comparative analysis. The benchmark is primarily contrasted only with NL4OPT, focusing on variable count. A more comprehensive validation is lacking, including:
    * Lack of Broader Comparison: It does not compare its scale (1,000 samples), problem type diversity, or variable distribution against other contemporary benchmarks in the field.
    * Incomplete Characterization: Key statistics that would allow researchers to fully understand the benchmark's profile and difficulty are missing or under-explained.
* **Failure to Cite and Discuss Key Related Work**: The paper overlooks critical and highly relevant recent research, creating a gap in its literature review and failing to properly situate its contribution. Notably absent are:
    * *Optibench Meets Resocratic: Measure and Improve LLMs for Optimization Modeling*. Zhicheng Yang, Yiwei Wang, Yinya Huang, Zhijiang Guo, Wei Shi, Xiongwei Han, Liang Feng, Linqi Song, Xiaodan Liang, Jing Tang. ICLR, 2025
    * *ORLM: A Customizable Framework in Training Large Models for Automated Optimization Modeling*. Chenyu Huang, Zhengyang Tang, Shixi Hu, Ruoqing Jiang, Xin Zheng, Dongdong Ge, Benyou Wang, Zizhuo Wang. Operations Research, 2025

**Questions:**

* Can you provide any qualitative analysis or examples where the sym prompt did successfully guide the LLM to a correct intermediate representation, even if it didn't always lead to executable code?
* You note that models were often "accurate" on infeasible problems, potentially because "it is infinitely easier to make a problem infeasible." Did you consider reporting accuracy scores separately for feasible and infeasible subsets of the benchmark?

If the authors address the weaknesses and answer my questions, I will be happy to raise the score.

---

### Official Review · Reviewer_hF9h · 2025-11-01

**Soundness:** 2
**Presentation:** 2
**Contribution:** 2
**Rating:** 4
**Confidence:** 4

**Summary:**

This paper proposes NLMOptimizer, a neurosymbolic framework aimed at addressing the challenge of generating operations research (OR) optimization problems from natural language descriptions. The main contributions include:

- Introducing the problem class for systematically generating symbolically represented optimization problems (e.g., Linear and Quadratic Programs) and constructing a dataset of 1000 problem instances covering integer, mixed-integer, and continuous types.
- Proposing the symInterchange class as an exploratory tool for mapping natural language problems into structured, solver-executable intermediate representations.
- Evaluating four instruction-tuned Large Language Models (LLaMa-3.3, LLaMa-4-Scout, Gemini-1.5-Pro, GPT-OSS-120B) on both the NLAOPT and NLMOptimizer datasets, revealing a significant performance drop on the more complex NLMOptimizer problems.
- Arguing that existing benchmarks (like NLAOPT) underestimate the task difficulty and advocating for the development of neurosymbolic methods that focus more on representational fidelity.

**Strengths:**

- The dataset is large-scale (1000 problems), covers multiple problem types (LP/QP/MILP, etc.), and solution validity is verified via Gurobi.
- The paper is logically structured, the experimental section is described in detail, and the appendix provides substantial theoretical and code support.
- Highlights critical deficiencies in current LLMs for OR problem representation, providing an important benchmark and direction for future research.

**Weaknesses:**

- Does not utilize the latest LLMs (e.g., GPT-5, Gemini-2.5-Pro, DeepSeek-R1), meaning the conclusions may not reflect the performance of current SOTA models.
- Lacks comparison with recent related work (e.g., IndustryOR, OptMATH), making it difficult to assess NLMOptimizer's relative advantages.
- SymInterchange class is only conceptually described, lacking concrete implementation, experimental validation, and a demonstration of its integration with the Problem class.
- While templates introduce variety, the naturalness or semantic consistency of the generated text is not evaluated.

**Questions:**

- Why were older models like Gemini-1.5-Pro chosen instead of the latest ones? Are there plans to re-evaluate performance on updated models?
- Could the paper provide detailed algorithms, code examples, or experimental results demonstrating the effectiveness of the SymInterchange class? How does it synergize with the Problem class?
- Are there plans to compare against recent benchmarks like IndustryOR or OptMATH? In which specific aspects does NLMOptimizer outperform these works?
- How was the realism and diversity of the generated natural language problems evaluated? Was semantic consistency considered beyond simple template filling?

---

### Official Review · Reviewer_wp6Y · 2025-11-05

**Soundness:** 2
**Presentation:** 2
**Contribution:** 2
**Rating:** 2
**Confidence:** 3

**Summary:**

In this paper, the authors proposed a new dataset of operations research (OR) problems, and demonstrated that it is challenging for several large language models to translate these problem descriptions into mathematical models.

The authors mentioned that they proposed a **SymInterchange** class to improve the accuracy of LLM formulation. However, I did not find sufficient details about this method and the results on the accuracy improvement.

I also felt that the authors may be unaware of more recent datasets and benchmarks that are more challenging than NL4OPT, which is the only existing dataset mentioned in the paper.

The description of the neurosymbolic framework has many mathematical notations and definitions without clear explanation or intuition. It is also not clear how it is related to the rest of the paper.

**Strengths:**

* The authors constructed a new dataset that is challenging for several LLMs under zero-shot prompting.

**Weaknesses:**

**Missing Recent Benchmarks and Datasets.** The authors exclusively mentioned NL4OPT in the paper, which was the first dataset for the task of translating OR problem description into mathematical models. However, there have been considerable efforts in developing more challenging benchmarks and datasets, such as [IndustryOR](https://huggingface.co/datasets/CardinalOperations/IndustryOR) and [OptMATH](https://openreview.net/forum?id=9P5e6iE4WK). These datasets are also challenging for various LLMs under zero-shot prompting and even after fine-tuning.

**Missing Recent Methods on Auto-Formulating OR Problems.** The authors evaluated existing LLMs under zero-shot prompting and "additional prompts" (without giving details on how additional prompts are constructed). There are advanced prompting methods tailored to auto-formulating OR problems (e.g., [Chain-of-Experts](https://openreview.net/forum?id=HobyL1B9CZ), [OptiMUS](https://proceedings.mlr.press/v235/ahmaditeshnizi24a.html), [a MCTS-based method](https://openreview.net/forum?id=33YrT1j0O0)), as well as fine-tuned LLMs for auto-formulating OR problems (e.g., [the fined-tuned LLM in OptMATH](https://openreview.net/forum?id=9P5e6iE4WK)). It is more reasonable to evaluate these methods on the proposed dataset.

**No Details on SymInterchange.** The abstract gave an impression that the authors proposed a new method, called **SymInterchange**, to improve the accuracy on their new dataset. However, there is barely any discussion on **SymInterchange**. In the main text, we are redirected to `Appendix C.3` for detail (`Line 1991`). But `Appendix C.3` has only three short paragraphs. There is also no experimental evaluation of **SymInterchange**.

**Unclear Description of the Neurosymbolic Framework.** There are heavy mathematical notations and definitions involved in the neurosymbolic framework.
* It is not clear how the math in `Page 4` is linked to the rest of the paper.
* It is not clear what is the purpose or the intuition of `Corollary 3.1`.
* It is not clear what the theorems in `Appendix B` are about. They all seem to be existing results.

**Questions:**

Please see my comments in "Weaknesses".

---

### Author Response · Authors · 2025-11-12
**The reviewers collectively misunderstood what was submitted, and expected methods, not a theoretically sound benchmark dataset for neurosymbolic learning.**

The reviewers collectively communicated that the empirical work presented is not sufficiently compelling for ICLR-26. Specifically, more recent models and approaches should be considered, so that if this work were included in ICLR-26, it would be frontier worthy. However, these reviews also did not content with our key argument, and in some cases, seemed to outright ignore it, focused as they were on the empirical performance of approaches that we contend are ultimately insufficient.

We argue in the paper, and ultimately advocate for a neurosymbolic approach that converts OR problems into terms drawn from E-Ring structures because this work is ultimately the sound theoretical basis for operations research problems. This is the content of Appendix B, which one of the reviewers (wp6y ) seemed confused about. We state multiple times throughout the paper that our choice of E-Ring formalism as the principle representation for OR optimization problems is pragmatically grounded, as it allows for a solution to the symbolic grounding problem in a way that presents an interchangeable formal object that represents a wide variety of operations research problems. We describe the formal mechanisms for why this is the case, and these reviewers seem incapable of understanding this presentation. Our empirical findings themselves showed that for simple problems, off the shelf models cannot perform well at these tasks. Of course 10 variables or less is not sufficiently representative of real life OR problems. But if models can't even faithfully convert those problems into correct symbolic representations, why should we expect they can do so with 100s or millions of variables. Such that their criticisms are salient, it would be worth investigating current SOTA models referenced in their reviews on these datasets.

Additionally, the reviewers seem properly confused about the work done in this paper. We introduce TWO classes, Problem, and SymInterchange. The majority of the work presented in this paper focused on the contributions of the Problem class, NOT the SymInterchange class. Our empirical contributions were a dataset that addresses deficiencies of other datasets, not other models or architectures, and not on the SymInterchange methods that would address this.

For this reason, we believe collectively that the reviewers were reviewing  a paper we did not write, and did not intend to submit. Their collective expectations demand something that was not provided -- we provided a sound basis for an alternative dataset, along with a mathematically informed justification for that basis, and not an architecture or prompt, or composition of models. Reviewers collectively decided that an applied model theoretic approach was in and of itself something not worth addressing, and rather expected more of the same from LLM models.

Finally, some reviewers nitpicking criticism seems to be finding flaws where none exist. LLM is capitalized properly throughout the paper, save for three instances. Two of those instances refer to function names, where LLM appears in substring forms as '_llm_', and once in the references, where following the bib style, only the first letter of the title is capitalized. To suggest, even obliquely, that we are in error here also suggests an inappropriate pedantry given the total failure to understand, or content with the actual arguments we presented in this paper.

---

### Note · Authors · 2025-11-12

**Comment:**

: The reviewers collectively communicated that the empirical work presented is not sufficiently compelling for ICLR-26. Specifically, more recent models and approaches should be considered, so that if this work were included in ICLR-26, it would be frontier worthy. However, these reviews also did not content with our key argument, and in some cases, seemed to outright ignore it, focused as they were on the empirical performance of approaches that we contend are ultimately insufficient.

We argue in the paper, and ultimately advocate for a neurosymbolic approach that converts OR problems into terms drawn from E-Ring structures because this work is ultimately the sound theoretical basis for operations research problems. This is the content of Appendix B, which one of the reviewers (wp6y ) seemed confused about. We state multiple times throughout the paper that our choice of E-Ring formalism as the principle representation for OR optimization problems is pragmatically grounded, as it allows for a solution to the symbolic grounding problem in a way that presents an interchangeable formal object that represents a wide variety of operations research problems. We describe the formal mechanisms for why this is the case, and these reviewers seem incapable of understanding this presentation. Our empirical findings themselves showed that for simple problems, off the shelf models cannot perform well at these tasks. Of course 10 variables or less is not sufficiently representative of real life OR problems. But if models can't even faithfully convert those problems into correct symbolic representations, why should we expect they can do so with 100s or millions of variables. Such that their criticisms are salient, it would be worth investigating current SOTA models referenced in their reviews on these datasets.

Additionally, the reviewers seem properly confused about the work done in this paper. We introduce TWO classes, Problem, and SymInterchange. The majority of the work presented in this paper focused on the contributions of the Problem class, NOT the SymInterchange class. Our empirical contributions were a dataset that addresses deficiencies of other datasets, not other models or architectures, and not on the SymInterchange methods that would address this.

For this reason, we believe collectively that the reviewers were reviewing a paper we did not write, and did not intend to submit. Their collective expectations demand something that was not provided -- we provided a sound basis for an alternative dataset, along with a mathematically informed justification for that basis, and not an architecture or prompt, or composition of models. Reviewers collectively decided that an applied model theoretic approach was in and of itself something not worth addressing, and rather expected more of the same from LLM models.

Finally, some reviewers nitpicking criticism seems to be finding flaws where none exist. LLM is capitalized properly throughout the paper, save for three instances. Two of those instances refer to function names, where LLM appears in substring forms as 'llm', and once in the references, where following the bib style, only the first letter of the title is capitalized. To suggest, even obliquely, that we are in error here also suggests an inappropriate pedantry given the total failure to understand, or content with the actual arguments we presented in this paper.

**Withdrawal Confirmation:**

I have read and agree with the venue's withdrawal policy on behalf of myself and my co-authors.